# Introduction to the NJIAS Himawari-8/9 cloud feature dataset for climate and typhoon research

Xiaoyong Zhuge[1], Xiaolei Zou[2], Lu Yu[1], Xin Li[1], Mingjian Zeng[1], Yilun Chen[3], Bing Zhang[1], Bin Yao[1], Fei Tang[1], Fengjiao Chen[1], Wanlin Kan[1]

[1]Key Laboratory of Transportation Meteorology of CMA, Nanjing Joint Institute for Atmospheric Sciences, Nanjing, 210041, China

[2]Joint Center of Data Assimilation for Research and Application, Nanjing University of Information Science and Technology, Nanjing, 210044, China

[3]School of Atmospheric Sciences, Sun Yat-sen University, Zhuhai, 519082, China

*Correspondence to*: Xiaolei Zou (xzou@nuist.edu.cn)

**Abstract.** The use of remote sensing method to accurately measure cloud properties and their spatiotemporal changes has been widely welcomed in many fields of atmospheric research. The Nanjing Joint Institute for Atmospheric Sciences (NJIAS) Himawari-8/9 Cloud Feature Dataset (HCFD) provides a comprehensive description of cloud features over the East Asia and west North Pacific regions for the 7 yr period from April 2016 to December 2022. Multiple cloud variables, such as cloud mask, phase/type, top height, optical thickness, and particle effective radius, as well as snow, dust and haze masks, were generated from the visible and infrared measurements of the Advanced Himawari Imager (AHI) onboard the Japanese geostationary satellites Himawari-8/9 using a series of cloud retrieval algorithms recently developed. Verifications with the *Cloud–Aerosol Lidar with Orthogonal Polarization* (CALIOP) 1-km cloud layer product and the *Moderate Resolution Imaging Spectroradiometer* (MODIS) Level-2 cloud product (MYD06) demonstrates that the NJIAS HCFD gives higher skill scores than the Japanese Himawari-8/9 operational cloud product for all cloud variables except for the particle effective radius. The NJIAS HCFD even outperforms the MYD06 in the nighttime cloud detection, cloud-top height/pressure/temperature estimation, and the infrared-only cloud-top phase determination. All evaluations are performed at the nominal 2 km scale, not including the effects of sub-pixel cloudiness or very thin cirrus. Two examples are presented, to demonstrate applications of the NJIAS HCFD for climate and typhoon research. The NJIAS HCFD has been published at the Science Data Bank (https://doi.org/10.57760/sciencedb.09950, Zhuge 2023a; https://doi.org/10.57760/sciencedb.09953, Zhuge 2023b; https://doi.org/10.57760/sciencedb.09954, Zhuge 2023c; https://doi.org/10.57760/sciencedb.10158, Zhuge 2023d; https://doi.org/10.57760/sciencedb.09945, Zhuge 2023e).

**1 Introduction**

Clouds play a crucial role in severe weather systems. The formation, development, and dissipation of convective storms are closely related to cloud microphysical processes (Zhuge and Zou, 2018; Liu et al., 2020). The intensity and size of a tropical cyclones are also indicated by the states of clouds (Zhuge et al., 2015; Sun et al., 2021). In addition, clouds modulate the planetary radiation budget by reflecting incoming solar radiation and absorbing outgoing long-wave radiation in Earth's climate system (Stephens, 2005; Yang et al., 2015) and affect the Earth's hydrological cycle by altering the water distribution through precipitation (Rosenfeld et al., 2014; Stevens and Bony, 2013). However, cloud processes are not yet well understood nor accurately predicted by current weather and climate models. Obtaining global cloud properties and their spatiotemporal changes has always been of great interest to weather and climate community at large.

Satellite remote sensing is an approach to observe and retrieve cloud properties on a global scale. There are two types of satellite sensors: active and passive sensors. Active sensors, such as the Cloud-Aerosol Lidar with Orthogonal Polarization (CALIOP) onboard the Cloud-Aerosol Lidar and Infrared Pathfinder Satellite Observation (CALIPSO) satellite (Winker et al., 2007), and the Cloud Profiling Radar (CPR) onboard the CloudSat satellite (Stephens et al., 2002), can provide cloud profile information at a high spatial resolution with high accuracy. However, these sensors have limited spatial coverage due to their nadir-only sampling mode. In contrast, the passive sensors provide measurements of wide swaths and multiple channels, which allows cloud top properties be retrieved over a large-coverage area. For example, the Moderate Resolution Imaging Spectroradiometer (MODIS) onboard the Earth Observing System *Aqua* and *Terra* platforms provide observations that are highly sensitive to cloud. It has 36 channels ranging from visible (VIS) to infrared (IR) at a nadir spatial resolution of 0.25–1 km (Platnick et al., 2003). The unique spectral and spatial capabilities have resulted in the generation of MODIS Level-2 cloud products (known as MOD06 for *Terra* and MYD06 for *Aqua*) which have been proven to have high accuracy and are widely used within the earth system science research community. Due to the safety concerns arising from MODIS extended service life, the National Aeronautics and Space Administration (NASA) is promoting a migration project to apply the MYD06 algorithms to the Visible Infrared Imaging

Radiometer Suite (VIIRS) onboard the U.S. polar-orbiting operational environmental satellites (Platnick et al., 2021). However, both MODIS and VIIRS have a revisit interval of 1-2 days, which means that the temporal evolution of clouds cannot be captured by these instruments.

The new generation of geostationary satellite imagers, such as the Advanced Himawari Imager (AHI) onboard Japanese Himawari-8/9 satellites (Bessho et al. 2016), the Advanced Baseline Imager

(ABI) onboard U.S. Geostationary Operational Environmental Satellite (GOES)-R series (Schmit et al., 2017), the Advanced Geostationary Radiation Imager (AGRI) onboard Chinese Fengyun-4 satellites (Yang et al., 2017), and the Flexible Combined Imager (FCI) onboard European Meteosat Third Generation (MTG; Holmlund et al. 2021), can continuously observe large-scale regions at a high spatiotemporal resolution. This capability enables a comprehensive remote sensing of various cloud

properties.

The GOES-R Algorithm Working Group has developed a series of retrieval algorithms for ABI cloud (Heidinger and Straka, 2013) and fog (Calvert and Pavolonis, 2010) masks, cloud height (Heidinger, 2012), cloud phase and type (Pavolonis, 2010), as well as daytime (Walther et al., 2013) and nighttime (Minnis and Heck, 2012) optical/microphysical parameters. For AHI operational cloud algorithms, the

techniques developed by Imai and Yoshida (2016) and Mouri et al. (2016a, b) are used for the AHI cloud mask, cloud height and cloud phase determinations, and a multifunctional algorithm called *Comprehensive Analysis Program for Cloud Optical Measurement* (CAPCOM) is employed to retrieve the optical and microphysical parameters for liquid-water (Nakajima and Nakajma, 1995; Kawamoto et al., 2001) and ice (Letu et al., 2019, 2020) clouds. The AHI level-2 operational cloud product from

September 2015 to the present at a low spatial resolution of $0.05\degree \times 0.05\degree$ is archived on the P-Tree System, Japan Aerospace Exploration Agency (JAXA). All cloud variables are available only during the daytime at solar zenith angles below $80\degree$. As a result, only the semi-diurnal variation of cloud cover (e.g., Shang et al., 2018; Yu et al., 2022) or convective activity (e.g., Li et al., 2021) during the daytime can be obtained from the AHI level-2 operational cloud product.

To supplement the JAXA operational cloud algorithms and products, starting from 2016, the authors have successfully developed multiple algorithms for AHI cloud mask (Zhuge and Zou, 2016; Zhuge et al., 2017), cloud-top phase (Zhuge et al., 2021a), cloud type (Zhang et al., 2019; Sun et al., 2019), and daytime cloud optical/microphysical parameters (DCOMPs; Zhuge et al., 2021b). They are now collectively referred to as Nanjing Joint Institute for Atmospheric Sciences (NJIAS) cloud retrieval

algorithms. Over the past three years, it has been discovered that the NJIAS cloud retrieval algorithms

have several shortcomings and weaknesses, such as inadequate detection of low-level clouds at high solar

zenith angles or over snow-covered surfaces, and insufficient masks of dust, haze and fog. Accordingly,

a number of enhancements to the NJIAS cloud retrieval algorithms have been implemented. Finally, 30

variables are generated at the 0.5 h interval in the 7 yr period from April 2016 to December 2022 using

these algorithms. They are named as the NJIAS Himawari-8/9 Cloud Feature Dataset (HCFD). The

objectives of this article are twofold: 1) to give an in-depth overview of the NJIAS HCFD, including the

updates made to NJIAS cloud retrieval algorithms since 2021; and 2) to objectively evaluate the accuracy

of NJIAS HCFD, particularly its comparative performance with existing datasets.

The remaining parts of this article are organized as follows. Section 2 gives a detailed overview of

the NJIAS HCFD. Section 3 presents results of an evaluation of the NJIAS HCFD accuracy against the

CALIOP and Collection-6.1 MYD06 datasets. Section 4 presents two application examples: one on cloud

climatology in southwestern China and the other on cloud and precipitation features of landfalling

typhoons. After a description on data availability (section 5), a summary and conclusions are given in

section 6.

## 2 Overview of the NJIAS HCFD

### 2.1 Input data

The primary sensor data employed by the NJIAS HCFD are the multispectral observations of the

AHI onboard Himawari-8/9. Himawari-8 became operational on July 7, 2015 and was replaced by its

successor, Himawari-9 on December 13, 2022. The AHI provides a full-disk scan every 10 min with a

spatial resolution of 0.5–2 km at the sub-satellite point around 140.7 °E. During the data dissemination

step, AHI full disk imagery is divided into ten segments from north to south by the Japan Meteorological

Agency. The NJIAS HCFD only focuses on Segments 2–4, covering the vast majority of the East Asia

and western North Pacific (WNP) regions. Given that the AHI IR channels have coarser spatial

resolutions (nominal 2 km) than the VIS and shortwave-IR (SWIR) ones (nominal 0.5–1 km), data from

finer-resolution channels are each aggregated to nominal 2 km resolution.

Clear-sky brightness temperatures (BTs) and transmission profiles for AHI 10 IR channels are

simulated by using the Community Radiative Transfer Model (CRTM) of version 2.2.3 (Han et al., 2007)

with the vertical profiles of pressure, temperature, water vapor and composition, as well as surface variables of surface skin temperature and 10-m wind, from the U.S. National Centers for Environmental

Prediction (NCEP) Final operational global (FNL) analyses (Kalnay et al., 1996) as the input. The NCEP FNL analysis, which has a $0.25° \times 0.25°$ horizontal resolution and a 6-h interval, is remapped to AHI observation times and pixels using a linear interpolation method. Other ancillary data including surface type, terrestrial elevation, and land surface emissivity are extracted from the one-minute land ecosystem classification product (http://modis-atmos.gsfc.nasa.gov/ECOSYSTEM/index.html), global 30 arc-

second elevation dataset (http://webmap.ornl.gov/ogcdown/dataset.jsp?ds_id510003), and University of Wisconsin–Madison High Spectral Resolution Emissivity dataset (http://cimss.ssec.wisc.edu/iremis), respectively.

**Table 1: List of output variables.**

| Short name | Long name | Assigned value or Unit |
|---|---|---|
| CldMask | Cloud mask | Confidently clear=0; Probably clear=1; Probably cloudy =2; Confidently cloudy=3 |
| FogMask | Fog/Low stratus mask | Probably Foggy = 1; Confidently foggy = 2 |
| CldType | Cloud type | Confidently clear=0; Probably clear=1; Broken=2; Warm water = 3; Supercooled water = 4; Mixed = 5; Opaque Ice = 6; Cirrus = 7; Overlapped = 8; Overshooting = 9 |
| CldType2 | Cloud type in ISCCP rule[1] | Confidently clear=0; Probably clear=1; Broken=2; Cu = 3; Sc = 4; St = 5; Ac = 6; As = 7; Ns = 8; Ci = 9; Cs=10; Cb=11 |
| CldPhase | Cloud-top thermodynamic phase | Clear =0; Warm-water = 1; Supercooled-water = 2; Mixed/uncertain = 3; Ice = 4 |
| CldTemperature | Cloud-top temperature | K |
| CldHeight | Cloud-top height | m AGL |
| CldPressure | Cloud-top pressure | hPa |
| ACHA_COD | Cloud optical thickness from the ACHA approach[2] | unitless |
| ACHA_CPS | Cloud-top particle effective radius from ACHA the approach[2] | μm |
| DCOMP*_COD[3] | Cloud optical thickness from the DCOMP approach[1] | unitless |
| DCOMP*_CPS[3] | Cloud-top particle effective radius from the DCOMP approach[1] | μm |
| DCOMP*_LWP[3] | Cloud liquid water path from the DCOMP approach[1] | g m$^{-2}$ |
| DCOMP*_IWP[3] | Cloud ice water path from the DCOMP approach[1] | g m$^{-2}$ |
| LatPC | Latitude after parallax corrections | °N |
| LonPC | Longitude after parallax corrections | °E |
| SST | Clear-sky sea skin temperature | K |
| ShadowMask | Shadow[1] | Shallow=1 |
| HazeMask | Haze[1] | Haze=1 |

| SnowMask | Snow and sea-ice surface[1] | Snow/Ice = 1; Permanent snow = 2 |
|----------|------------------------------|----------------------------------|
| FireMask | Active fire | Possible fire=1; Confident fire=2 |
| DustMask | Dust | Possible dust=1; Confident dust=2 |

[1] Daytime only.

[2] Only reliable for cirrus clouds.

[3] DCOMP* represents DCOMP35, DCOMP36 and DCOMP37, meaning the variables are derived using 0.64-μm and either 1.6- , 2.3-, or 3.9-μm channels, respectively.

## 2.2 Output variables

The NJIAS HCFD provides a comprehensive description of cloud features over the East Asia and WNP regions. It includes 30 variables, such as cloud mask, cloud optical thickness (τ), cloud-top thermodynamic phase, cloud-top height (CTH), and cloud-top particle effective radius (Re), as well as snow, dust and haze masks. The 30 output variables are briefly described in Table 1.

## 2.3 NJIAS cloud retrieval algorithms

During the past three years, a number of improvements to the NJIAS cloud retrieval algorithms have been incorporated. Improvements include the following.

### 2.3.1 Cloud mask algorithm refinements

The NJIAS cloud mask algorithm is developed on the basis of previous two works (Zhuge and Zou, 2016; Zhuge et al., 2017). Eight of ten cloud-mask tests used in Zhuge and Zou (2016) and one test used in Zhuge et al. (2017) are inherited. These nine cloud-mask tests are called relative thermal contrast test (RTCT), emissivity at tropopause test (ETROP), positive channel-14 minus 15 test (PFMFT), relative channel-14 minus 15 test (RFMFT), cirrus water vapor test (CIRH2O), uniform low stratus test (ULST), new optically thin cloud test (N-OTC), temporal IR test (TEMPIR), and VIS-based cloud index test (VCI). To enhance the detection of low-level clouds, additional six cloud-mask tests are employed by the NJIAS algorithm, that is, relative VIS contrast test (RVCT), reflectance ratio test (RRT), terminator thermal stability test (TTST), nighttime low stratus test over desert (DZT_NLS), daytime low stratus test over sunglint regions (SG_DLS), and reflectance similarity test (RST). The mathematical formulas for the above-mentioned 15 cloud-mask tests are listed in Table 2. Note that $O_{x\mu m}$ is the observed BT or reflectance at x-μm wavelength, $B_{x\mu m}$ is the simulated x-μm BT under clear-sky conditions, $I_{x\mu m}\left(T\right)$

represents the radiance at temperature $T$ and x-µm wavelength that is computed by the Planck function. The threshold ( $\varepsilon$ ) for a certain test is generally derived via a comparison of co-located AHI/ABI with CALIOP data (Zhuge and Zou, 2016; Zhuge et al., 2017). The flowchart of the NJIAS cloud mask algorithm is shown in Fig. 1.


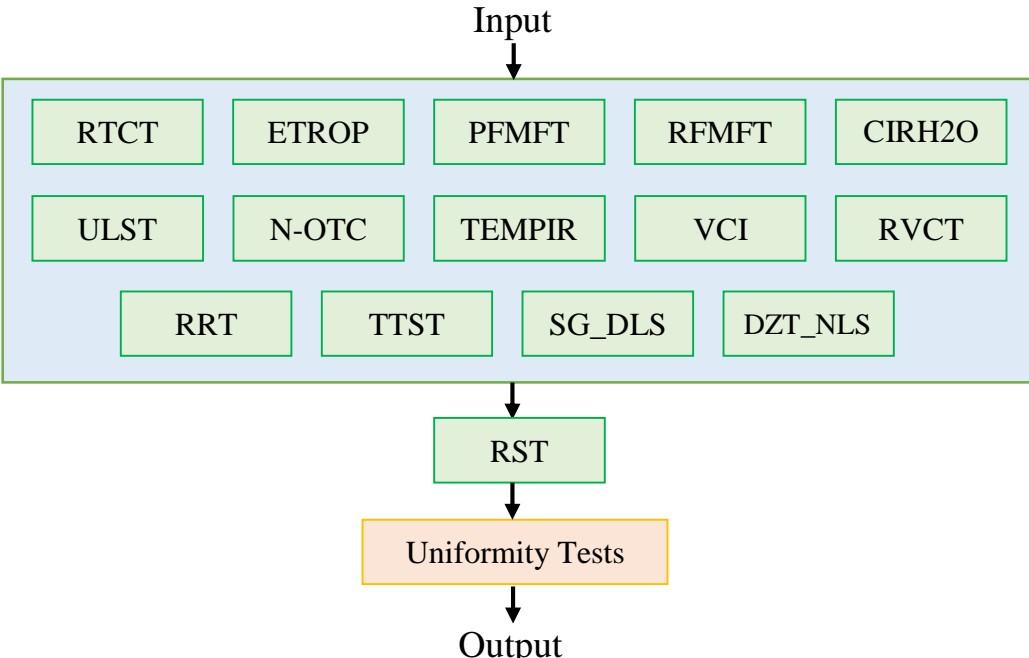

**Figure 1: Flowchart of the NJIAS cloud mask algorithm.**

**Table 2: Names and mathematical formulas for the 15 tests employed by the NJIAS cloud mask algorithm.**

| Name | Condition for cloudy pixels | Remarks |
|---|---|---|
| RTCT | $\left(O_{11.2\mu m}^{\max} - O_{11.2\mu m}\right) > \varepsilon_{RTCT}$ | Inherited from Zhuge and Zou (2016) |
| ETROP | $\dfrac{I_{11.2\mu m}\left(O_{11.2\mu m}\right) - I_{11.2\mu m}\left(B_{11.2\mu m}\right)}{R_{11.2\mu m}^{trop} - I_{11.2\mu m}\left(B_{11.2\mu m}\right)} > \varepsilon_{ETROP}$ | |
| PFMFT | $\left(O_{11.2\mu m} - O_{12.4\mu m}\right) - \left(B_{11.2\mu m} - B_{12.4\mu m}\right)\cdot\dfrac{\left(O_{11.2\mu m} - 260K\right)}{\left(B_{12.4\mu m} - 260K\right)} > \varepsilon_{PFMFT}$ | |
| RFMFT | $\left|\left(O_{11.2\mu m} - O_{12.4\mu m}\right) - \left(O_{11.2\mu m}^{NWC} - O_{12.4\mu m}^{NWC}\right)\right| > \varepsilon_{RFMFT}$ | |
| CIRH2O | $\rho\left(O_{11.2\mu m}, O_{7.3\mu m}\right) > \varepsilon_{CIRH2O}$ | |
| ULST | $\dfrac{I_{3.9\mu m}\left(B_{3.9\mu m}\right)}{I_{3.9\mu m}\left(B_{11.2\mu m}\right)} - \dfrac{I_{3.9\mu m}\left(O_{3.9\mu m}\right)}{I_{3.9\mu m}\left(O_{11.2\mu m}\right)} > \varepsilon_{ULST}$ | |
| N-OTC | $O_{3.9\mu m} - O_{12.4\mu m} > \varepsilon_{N-OTC}$ | |
| TEMPIR | $O_{11.2\mu m}^{-10\min} - O_{11.2\mu m} > \varepsilon_{TEMPIR}$ | |
| VCI | $\sqrt{\dfrac{\left(O_{0.47\mu m}^{'} - O_{0.64\mu m}^{'}\right)^2 + \left(O_{0.47\mu m}^{'} - O_{0.86\mu m}^{'}\right)^2 + \left(O_{0.64\mu m}^{'} - O_{0.86\mu m}^{'}\right)^2}{3}} < \varepsilon_{VCI}$ | Inherited from Zhuge et al. (2017) |
| RVCT | $O_{0.64\mu m}^{Norm} - O_{0.64\mu m}^{Norm,\min} > \varepsilon_{M-RVCT}$ | Adopted from Heidinger and Straka (2013) |
| RRT | $\dfrac{O_{0.86\mu m}}{O_{0.64\mu m}} > \varepsilon_{RRT}$ | |
| TTST | $\left|O_{11.2\mu m}^{-1hr} - O_{11.2\mu m}\right| < 2$ and $CM^{-1hr} = TRUE$ and $\left|\left(O_{11.2\mu m}^{-1hr} - O_{8.6\mu m}^{-1hr}\right) - \left(O_{11.2\mu m} - O_{8.6\mu m}\right)\right| < \varepsilon_{TTST}$ | |
| SG_DLS | $B_{3.9\mu m} - O_{3.9\mu m} > \varepsilon_{SG\_DLS1}$ or $\dfrac{\left(O_{3.9\mu m} - O_{10.4\mu m}\right)}{O_{0.64\mu m}} < \varepsilon_{SG\_DLS2}$ | Newly added |
| DZT_NLS | $O_{12.4\mu m} - O_{10.4\mu m} < 0$ and $\left(O_{10.4\mu m} - O_{3.9\mu m} + 5\right)/10 - \left(O_{12.4\mu m} - O_{10.4\mu m} + 4\right)/6 > 0.16$ and $\dfrac{I_{3.9\mu m}\left(B_{3.9\mu m}\right)}{I_{3.9\mu m}\left(B_{11.2\mu m}\right)} - \dfrac{I_{3.9\mu m}\left(O_{3.9\mu m}\right)}{I_{3.9\mu m}\left(O_{11.2\mu m}\right)} > \varepsilon_{DZT\_NLS}$ | |
| RST | $\dfrac{O_{1.6\mu m}}{O_{0.64\mu m}} > 0.8$ and $O_{1.6\mu m}^{Norm} > \dfrac{\theta_{sol}}{300} - 0.05$ and $CM^{Neighbor} = TRUE$ and $\dfrac{O_{0.64\mu m}}{O_{0.64\mu m}^{Neighbor}} > \varepsilon_{RST}$ | |


For cloud detection over sun-glint regions, SG_DLS assumes that sea surface reflectance is greater than that of clouds. Thus, the 3.9-µm BTs over cloudy areas should be lower than those of model

simulations under clear-sky conditions. SG_DLS also compares the reflectance between 3.9-μm and

0.64-μm channels by simply using the formula $\dfrac{\left(O_{3.9\,\mu m}-O_{10.4\,\mu m}\right)}{O_{0.64\,\mu m}}$ and marks those pixels as cloudy

where the reflectance in the 3.9-μm channel is significantly weaker compared to that in the 0.64-μm

channel. During nighttime, the low-level clouds and clear-sky desert have very similar characteristics of

3.9-μm emissivity. Relative to ULST, DZT_NLS employs two extra criteria ($O_{12.4\,\mu m}-O_{10.4\,\mu m}<0$ and

$\left(O_{10.4\,\mu m}-O_{3.9\,\mu m}+5\right)/10-\left(O_{12.4\,\mu m}-O_{10.4\,\mu m}+4\right)/6>0.16$) so that the clear-sky desert pixels would

not be falsely flagged as cloudy.

175       Detection of low-level clouds at high solar zenith angles is challenging since the VIS reflectance

becomes very sensitive to aerosol and noise. To mitigate the labeling of haze pixels as being cloudy, VCI

and RRT were slightly modified. The pixels should firstly satisfy two basic conditions

($O_{1.6\,\mu m}^{Norm}>\dfrac{\theta_{sol}}{300}-0.05$ and $323-O_{11.2\,\mu m}>150\cdot O_{1.6\,\mu m}^{Norm}$) before they could proceed to next step.

Here, $\theta_{sol}$ is the solar zenith angle in degree, and $O_{1.6\,\mu m}^{Norm}$ is the 1.6-μm reflectance normalized by the

cosine of $\theta_{sol}$. Meanwhile, given that existing three reflectance-based tests (i.e., VCI, RVCT, and RRT)

are not as effective as at noon, TTST and RST are incorporated into the NJIAS cloud mask algorithm to

improve cloud detection at high solar zenith angles. As described by Heidinger and Straka (2013), TTST

classifies a pixel as cloudy if its IR spectral signatures are similar to those of a cloudy pixel that was

detected at the same location one hour ago. RST is a completely new cloud-mask test, being specifically

utilized for pixels with a solar zenith angle between 60 °and 83 °. The RST is implemented subsequent to

the preliminary cloud mask determination derived from the other 14 tests. The objective of RST is to

spatially extend the initial cloud "seeds" to their neighboring pixels that exhibit similar reflectance

characteristics. Again, these candidate cloudy pixels should firstly satisfy non-haze conditions

($\dfrac{O_{1.6\,\mu m}}{O_{0.64\,\mu m}}>0.8$ and $O_{1.6\,\mu m}^{Norm}>\dfrac{\theta_{sol}}{300}-0.05$). Figure 2 illustrates the utility of incorporating the RST

for low-level cloud detection in the early morning. The scene occurred at 23:00 UTC 10 April 2023,

when a vast expanse of quasi-stationary cloud belts were located over southern China. When detecting

clouds without RST, a lot of foggy and/or stratus pixels were missed, and thus the identified cloud belts

were fragmented (Fig. 2c). Cloud mask results with RST are much more reasonable (Fig. 2d).

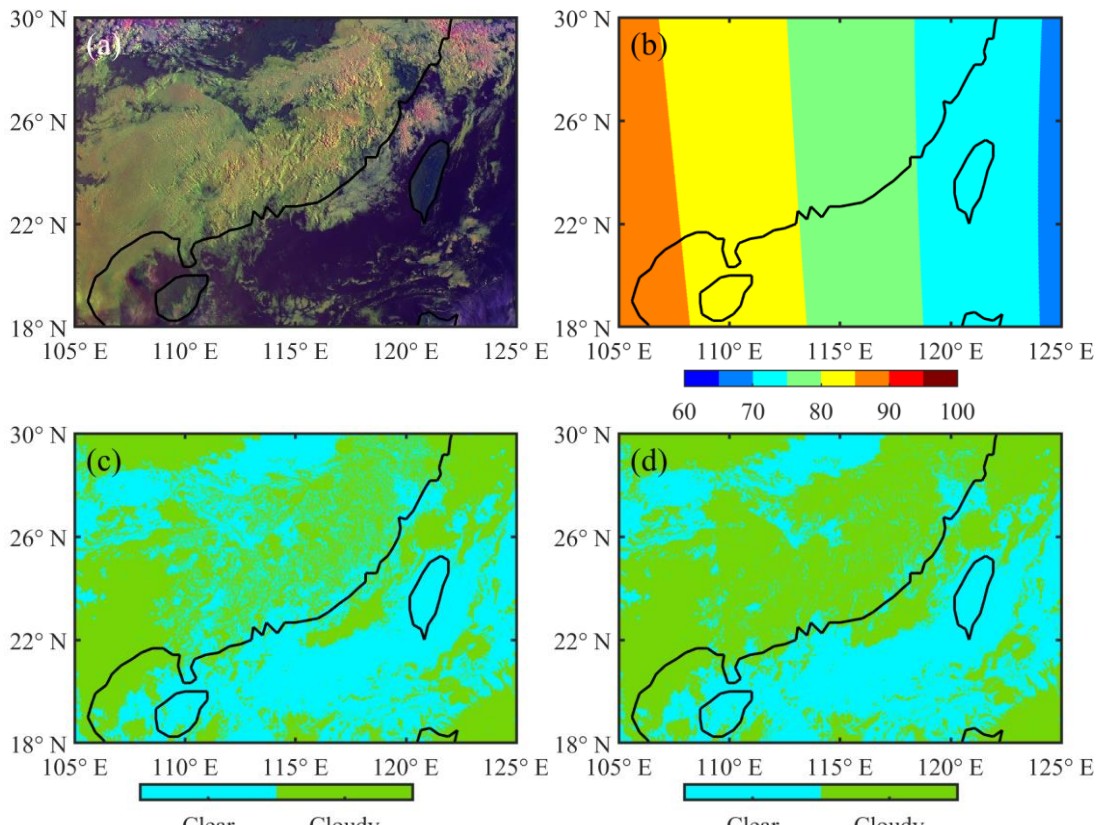

**Figure 2: (a) AHI false-color image (red, 0.64 µm; green, 1.6 µm; blue, 11.2 µm reversed) showing land/ocean in dark, thick ice clouds in magenta, cirrus in blue, and low clouds in yellow or white, (b) solar zenith angle (unit: degree), and (c)-(d) cloud mask results (c) without and (d) with RST at 23:00 UTC on 10 April 2023.**

Like other cloud mask algorithms, the NJIAS algorithm also generates a four-level mask whose categories are confidently clear, probably clear, probably cloudy, and confidently cloudy. Probably clear pixels are defined as those failing the uniformity tests, and probably cloudy pixels are those located at cloud edges.

### 2.3.2 Newly added snow, dust, and haze mask algorithms

Snow mask is an important procedure implemented before cloud mask. In the NJIAS algorithm, the pixels satisfy one of following three conditions are firstly identified as snow candidates: (1) they are over oceans with surface temperature analyses being lower than 263 K, (2) the underlying surface type is "permanent snow", and (3) both the normalized differential snow index (NDSI; $\frac{O_{0.64\mu m} - O_{1.6\mu m}}{O_{0.64\mu m} + O_{1.6\mu m}}$) and

the enhanced NDSI ($\frac{O_{0.64\mu m}^{0.33} - O_{1.6\mu m}^{0.33}}{O_{0.64\mu m}^{0.33} + O_{1.6\mu m}^{0.33}}$) are larger than 0.1, and meanwhile the normalized differential

vegetation index ($\frac{O_{0.86\mu m} - O_{0.64\mu m}}{O_{0.86\mu m} + O_{0.64\mu m}}$) is larger than -0.1. A serious of strict tests are then performed to

rule out the candidates presenting unique spectral characteristics of ice clouds (e.g., mobile; more

apparent on the water vapor images; much colder than the surface). However, the pixels that have an

NDSI value greater than 0.1 and were classified as snow one hour ago would be restored to snow again.

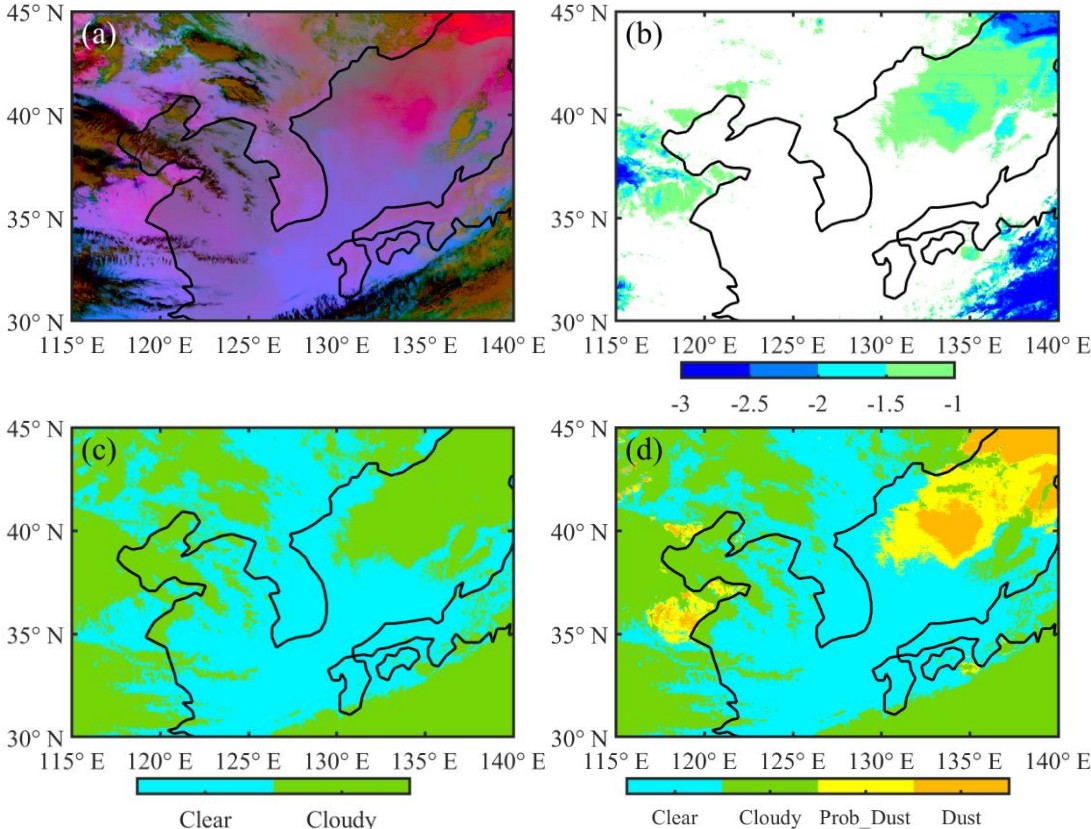

**Figure 3: (a) AHI "Dust" RGB composite image ( dust in pinkish color) and (b) NFMFT value (unit: K), and**

**(c)-(d) cloud mask results derived from (c) old and (d) new versions of the NJIAS cloud algorithms at 09:00**

**UTC on 12 April 2023.**

In the old version of NJIAS cloud mask algorithm, dust was often identified as cloudy, especially

when it is transported over oceans. A remarkable example of this occurred at 09:00 UTC 12 April 2023

(Fig. 3). The poor performance is primarily a result of the usage of the negative channel-14 minus 15 test

(NFMFT) that was originally applied to detect opaque clouds. In fact, the dust can generate a NFMFT

value ($\left(O_{11.2\mu m} - O_{12.4\mu m}\right) - \left(B_{11.2\mu m} - B_{12.4\mu m}\right)$ ) as great as the opaque clouds, as shown in Fig. 3b. Now, NFMFT is removed from the NJIAS cloud mask algorithm but added to the NJIAS dust mask algorithm, which originally included an empirically dust mask test developed based on the principle used by "Dust" RGB composite images (Lensky and Rosenfeld, 2008). The dust mask is implemented after cloud mask. Accordingly, cloud mask results derived from the NJIAS cloud algorithms are improved (Fig. 3d).

Similar consideration is applied to haze detection. The reflectance gross contrast test (RGCT) that was employed by various cloud mask algorithms is added to the haze mask algorithm. RGCT works on the assumption that clouds have larger 0.64-μm reflectance than clear sky, which is also true for haze. The original haze mask algorithm only included a heavy aerosol test—Test 1 in Hutchison et al. (2008), assuming that haze is transparent at the 2.3-μm wavelength but much reflective at the 0.64-μm wavelength.

### 2.3.3 Updates to the cloud-top property algorithm

The NJIAS cloud height algorithm follows mainly the architecture of the ABI Cloud Height Algorithm (ACHA; Heidinger, 2012). It derives cloud-top temperature (CTT), CTH, cloud-top pressure (CTP), $\tau$, and Re with a consistent accuracy for day and night. Note that $\tau$ and Re from the ACHA approach are only reliable for cirrus clouds because the long-wave IR observations cannot provide the desired sensitivity to cloud microphysics for optically thick clouds. Besides, the CTH in the NJIAS algorithm is measured above ground level (AGL), i.e., true altitude minus terrain elevation, which is different from the definition used in the MYD06 algorithm and the ACHA.

The NJIAS IR cloud-top phase algorithm is developed based on Zhuge et al. (2021a). It categorizes cloudy tops into liquid-water, ice, and mixed/uncertain phases, by employing the IR-window and IR-water vapor channels as well as several spectral and spatial tests. The liquid-water phase is further refined into being either supercooled-water or warm-water, depending on whether the CTT is below 0 ℃ or not. For ice-phase cloud tops, they are further divided into opaque-ice, cirrus, overlapped, and overshooting tops based on the results of the BT-based cirrus test, a beta-parameter-based overlap test, and a cloud-emissivity-based overshooting test (Platnick et al., 2019). In addition, a new cloud type named "broken" is defined for cirrus pixels which are located at cloud edges (i.e., cloud-mask value equals 2).

A pixel will be identified as probably foggy if it is liquid-water phase and the spatial uniformity

(i.e., the standard deviation of 11.2-μm BTs) over a 3×3 pixels array is below 0.5 K. Meanwhile, the 11.2-μm BT difference between satellite observations and model simulations (OMB) should be less negative than -10 (-15) K over land during daytime (nighttime) or -6 K over oceans all day. Subsequently, confidently foggy pixels are determined from the probably foggy ones if they have been classified as confidently cloudy and their spatial uniformity are below 0.3 K.

### 2.3.4 Updates to the DCOMP algorithm

Same as Zhuge et al. (2021b), the NJIAS DCOMP algorithm uses the bispectral method described by Nakajima and King (1990) in the daytime $\tau$ and Re retrievals. Three pairs of non-absorption and water-absorption channels at VIS, SWIR, and mid-wave IR wavelengths are employed to separately derive three DCOMP products (designated as DCOMP35, DCOMP36 and DCOMP37, meaning a combination of 0.64-μm and either 1.6-, 2.3-, or 3.9-μm channels, respectively). The NJIAS DCOMP algorithm utilizes parameterization schemes and retrieval procedures that are nearly consistent to those used in Zhuge et al. (2021b) except for the lookup tables (LUTs).

**Table 3: Grid point values of the LUT parameters.**

| Parameter | Number of points | Grid point values |
|---|---|---|
| Re (μm) | 16 | 3, 4, 5, 6, 7, 8, 9, 10, 12, 14, 16, 18, 20, 22, 24, 25 (liquid-water cloud) |
|  | 12 | 5, 10, 15, 20, 25, 30, 35, 40, 45, 50, 55, 60 (ice cloud) |
| $\tau$ | 34 | 0.05, 0.10, 0.25, 0.5, 0.75, 1.0, 1.25, 1.5, 1.75, 2.0, 2.39, 2.87, 3.45, 4.14, 4.97, 6.0, 7.15, 8.58, 10.30, 12.36, 14.83, 17.80, 21.36, 25.63, 30.76, 36.91, 44.30, 53.16, 63.80, 76.56, 91.88, 110.26, 132.31, 158.78 |
| $\mu_{sat}$ | 28 | 0.40, 0.45, 0.50, 0.55, 0.60, 0.65, 0.70, 0.75, 0.7625, 0.7750, 0.7875, 0.8000, 0.8125, 0.8250, 0.8375, 0.8500, 0.8625, 0.8750, 0.8875, 0.900, 0.9125, 0.9250, 0.9375, 0.9500, 0.9625, 0.9750, 0.9875, 1.0 |
| $\mu_{sol}$ | 33 | 0.15, 0.20, 0.25, 0.30, 0.35, 0.40, 0.45, 0.50, 0.55, 0.60, 0.65, 0.70, 0.75, 0.7625, 0.7750, 0.7875, 0.8000, 0.8125, 0.8250, 0.8375, 0.8500, 0.8625, 0.8750, 0.8875, 0.900, 0.9125, 0.9250, 0.9375, 0.9500, 0.9625, 0.9750, 0.9875, 1.0 |
| $\Delta\varphi$ (°) | 37 | 0:5:180 |

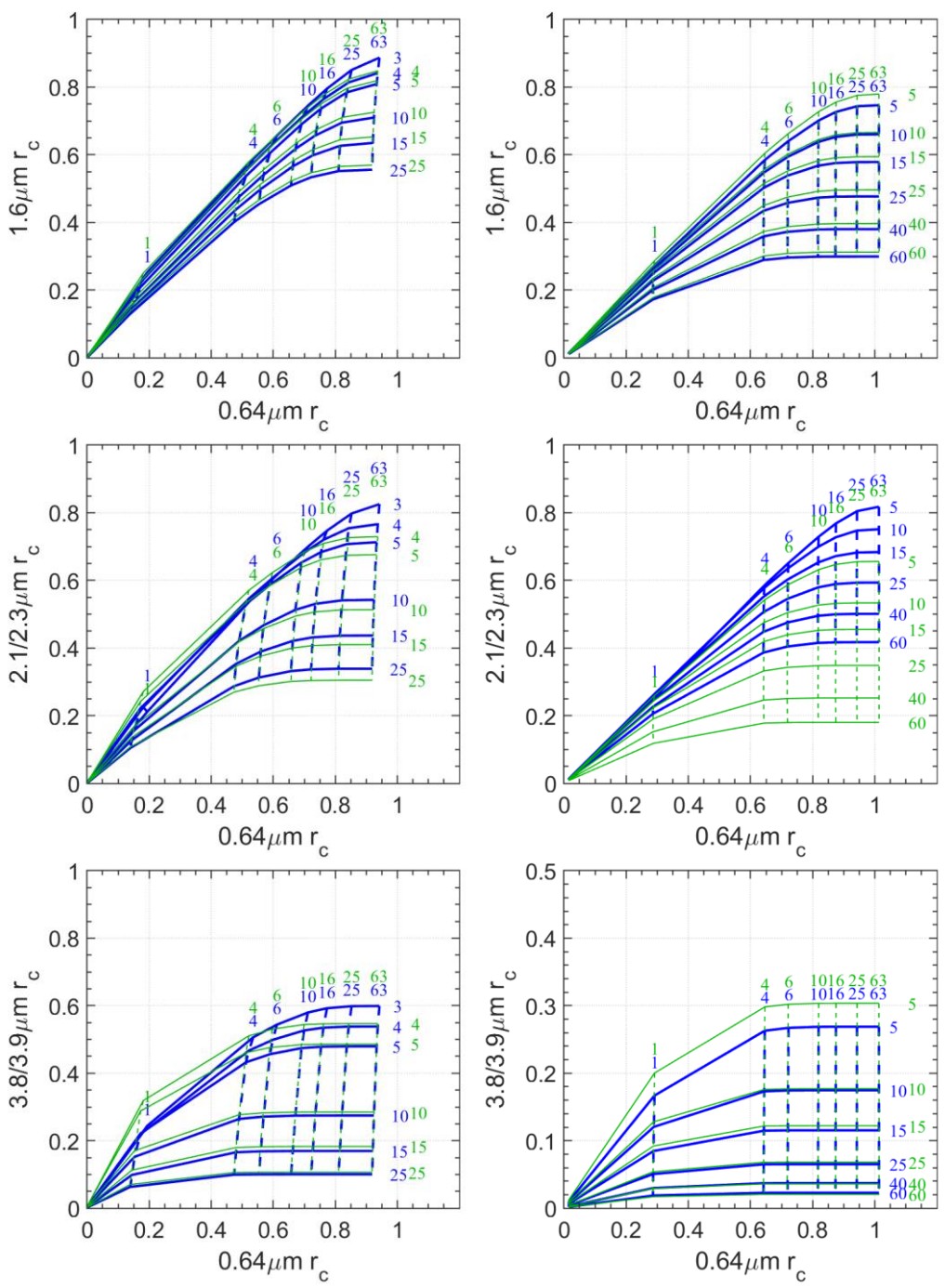

275

**Figure 4: Variations of $r_c$ at 0.64 µm and either 1.6 (top panels), ~2.2 (middle panels), or ~3.8 µm (bottom panels) for $Re$ = 3, 4, 5, 10, 15 and 25 µm (solid curve) and $\tau$ =1, 4, 6, 10, 16, 25 and 63 (dashed curve) for liquid-water phase (left panels) and for $Re$ = 5, 10, 15, 25, 40 and 60 µm (solid curve) and $\tau$ =1, 4, 6, 10, 16, 25 and 63 (dashed curve) for ice phase (right panels) from Collection-6.1 MYD06 (green) and NJIAS (blue)**

280 **datasets when $\mu_{sol} = \mu_{sat}$ = 0.5 and $\Delta\varphi$ = 60 °.**

Forward radiative transfer calculations for the LUTs were performed with the discrete ordinates

radiative transfer (DISORT) model implemented in libRadTran 2.0.3 (Mayer and Kylling, 2005; Emde et al., 2016). The atmospheric temperature and humidity profile is the U.S. Standard Atmosphere, and the absorption /scattering by air molecules or aerosols are neglected. The cloud layer is assumed to be 1 km thick and placed at an altitude of 5 km above a non-reflecting surface. The bulk single-scattering properties of clouds are considered separately for liquid-water and ice clouds. For liquid-water clouds, the scattering properties of water droplets are computed from Lorenz–Mie theory, assuming a gamma size distribution. For ice clouds, a scattering parameterization named Baum_v36 (Heymsfield et al., 2013; Yang et al., 2013; Baum et al., 2014) with ice crystal habit of severely roughened aggregated column is used. The water droplet and ice crystal assumptions are identical to those in the Collection-6.1 MYD06 algorithm. The final LUTs of cloud emissivity, reflectance, and transmissions as well as the spherical albedo are functions of Re, τ, the cosine of satellite zenith angle ( $\mu_{sat}$ ), the cosine of solar zenith angle ( $\mu_{sol}$ ), and the relative azimuth angle ( $\Delta\varphi$ ). Table 3 summarizes the grid point values for Re, τ, $\mu_{sat}$, $\mu_{sol}$ and $\Delta\varphi$ used in constructing the LUTs. Figure 4 shows visualizations of cloud reflectance ( $r_c$ ) at 0.64 μm and either 1.6, ~2.2, or ~3.8 μm for liquid-water and ice clouds for an arbitrarily chosen solar-viewing geometry. Green and blue curves are the LUTs used by Collection-6.1 MYD06 and NJIAS algorithms, respectively. Relative to the pairs of 0.64–1.6-μm channels and 0.64–~3.8-μm channels, the pair of 0.64–~2.2-μm channels has a noticeable difference in the LUTs of $r_c$ between MYD06 and NJIAS algorithms. The 2.3-μm $r_c$ values of the NJIAS LUTs are systematically larger than the 2.1-μm $r_c$ values of the MYD06 LUTs when the τ, Re, and solar-viewing geometry are same. This characteristic is especially significant for ice clouds.

Once τ and Re are determined, these two retrievals are used subsequently to calculate the total mass of water in a cloud column, known as liquid water path (LWP) and ice water path (IWP) for liquid-water and ice clouds, respectively. Assuming a vertical homogeneity of cloud, LWP (IWP) is derived using $\frac{4\rho}{3Q_e}R_e\tau$ (Stephens, 1978; Khanal and Wang, 2018), where $\rho$ is the density of liquid water (ice), and $Q_e$ is the liquid water (ice) extinction efficiency. Meanwhile, the CTP and τ retrievals are applied for determining cloud types based on the International Satellite Cloud Climatology Project (ISCCP) rule

(Rossow and Schiffer, 1999).

## 2.4 Cloud products

Currently, the NJIAS HCFD has three cloud products, namely FLDK (for Segments 2–4 of the full disk imagery), 0.04Deg (on regular latitude-longitude grids at 0.04 ° × 0.04 ° resolution) and TyWNP (for WNP Typhoons). The 0.04Deg and TyWNP products can be directly derived from the FLDK product via projection conversion using the nearest-neighbor approach. For the TyWNP product, typhoon center positions are determined by the tropical-cyclone-red-green-blue (TC-RGB) composites, as introduced in Chen et al. (2022). Table 4 lists the coverage and resolution in space and time for two products. A finer resolution would retain more clouds of ~2 km size. Users can select any of the three cloud products appropriate for their purpose.

**Table 4: Brief descriptions of three products of the NJIAS HCFD.**

| Product Name | Variables Included | Domain Coverage | Time Period | Spatial Resolution | Time Interval |
|---|---|---|---|---|---|
| FLDK | all variables | Segments 2–4 of the Himawari-8/9 full disk imagery | April 2016–December 2022 | 2 km at the sub-satellite point | |
| 0.04Deg | all variables except ShadowMask, HazeMask, FireMask, SST | 50 °N–10 °N, 90 °E–170 °W | April 2016–December 2022 | 0.04 ° | 30 minutes |
| TyWNP | all variables except ShadowMask, SnowMask, DustMask, HazeMask, FireMask, SST | a 20 ° ×20 ° longitude-latitude grid box surrounding the typhoon center | typhoon seasons from 2016 to 2022 | 0.02 ° | |

## 3 Evaluation of the NJIAS HCFD

In this section, results obtained by the NJIAS cloud mask and cloud-top property algorithms are objectively evaluated at the nominal 2-km pixel level against the CALIOP 1-km cloud layer products of version 4.20 (Avery et al., 2020) in the whole year of 2017. Because the CALIOP and AHI operate under different sampling schemes, only those AHI pixels within which the CALIOP cloud identification results are in complete agreement are retained. The temporal difference between CALIOP and AHI observations is limited to ±5 min. Also evaluated against CALIOP data are the Collection-6.1 MYD06 and JAXA

cloud products to make a comparison on the performance of NJIAS HCFD with these two existing cloud

feature datasets. The values at the MYD06/JAXA grids that are spatiotemporally nearest to the CALIOP

columns are used.

Collection-6.1 MYD06 dataset is employed to evaluate the NJIAS DCOMP retrievals. Similar to

the collocation between CALIOP and AHI pixels, all of the MODIS pixels within one AHI pixel shall

have a consistent phase, otherwise this MODIS-AHI data pair will not be included. For those pairs that

are retained, the retrievals of MODIS pixels within each matched AHI pixel are averaged first before the

comparison with the AHI retrievals.

**3.1 Cloud mask results**

The CALIOP columns with zero cloud layer are assigned to clear-sky category, and those with at

least one cloud layer are assigned to cloudy category. The CALIOP columns are then aggregated to

completely cloudy, completely clear-sky, and sub-pixel cloudy cases at nominal 2 km scales. Figure 5

shows the proportions of confidently clear, probably clear, probably cloudy and confidently cloudy pixels

in MYD06, NJIAS and JAXA cloud-mask results for three types of CALIOP cases. It is noted that the

JAXA product has the largest proportions of probably cloudy and the smallest proportions of probably

clear pixels among three cloud products. Overall, the MYD06 classifications are in best agreement with

those of CALIOP with higher confidence during daytime. The NJIAS classification results are similar to

the MODIS results with fractional differences of less than 10%. Three products (MYD06, NJIAS and

JAXA) have a probability of 25–35% to classify sub-pixel cloudy cases as confidently clear or probably

clear over oceans or during daytime. This probability increases to approximately 47% for the NJIAS

product over continental areas at night.


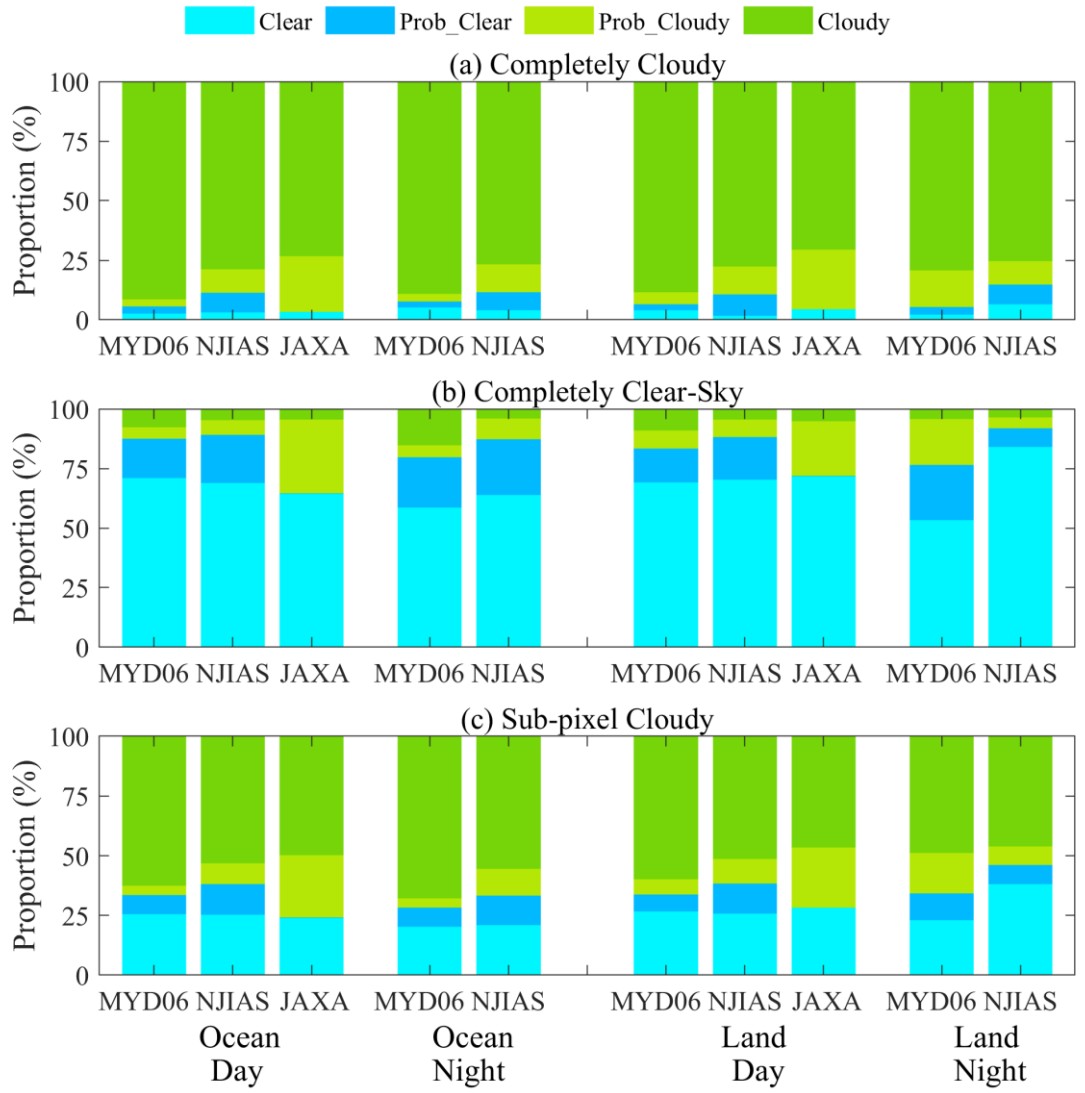

**Figure 5: Proportions of confidently clear, probably clear, probably cloudy, and confidently cloudy pixels in MYD06, NJIAS and JAXA cloud-mask results for CALIOP-observed (a) completely cloudy, (b) completely clear-sky, and (c) sub-pixel cloudy cases in 2017.**


To quantitatively evaluate the cloud-mask retrievals, the following four indices are introduced: probability of detection (POD), false-alarm rate (FAR), Heike skill score (HSS), and the equitable threat score (ETS). The definitions of the POD, FAR, HSS and ETS were described in Zhuge et al. (2011). Table 5 lists the scores of POD, FAR, HSS and ETS for cloud-mask retrievals of three datasets. Here,

confidently cloudy and probably cloudy are grouped as "cloudy" while confidently clear and probably clear are grouped as "clear". It can be seen that MYD06 and JAXA datasets always have a POD greater than 92%, regardless over oceans or land. The MYD06 also has a low FAR for all scenarios except during

nighttime over land. In contrast, the JAXA dataset has high FARs of more than 12% over oceans and land. The PODs and FARs for the NJIAS algorithm are ~88% and ~6%, respectively. Consequently, the

NJIAS HCFD achieves an HSS of 0.75 and an ETS of 0.60 during nighttime over land, surpassing the MYD06 dataset which has an HSS of 0.73 and an ETS of 0.57. NJIAS HCFD and MYD06 datasets have same skill scores of HSS (0.72) and ETS (0.56) during nighttime over oceans. In daytime scenarios, the NJIAS HCFD outperforms the JAXA dataset, but not exceeding the MYD06. Note that the aforementioned statistical analysis excluded all cases with sub-pixel cloudiness or very thin cirrus

(Karlsson et al., 2018; 2023). If the sub-pixel cloudy cases were misinterpreted as either completely clear-sky or completely cloudy, the estimation of all the scores would be biased unpredictably.

**Table 5: Sample sizes and POD, FAR, HSS and ETS sores for cloud-mask retrievals of MYD06, NJIAS and JAXA datasets over oceans and land and during daytime and nighttime when validated with CALIOP**

**products in the whole year of 2017. The highest skill scores for each scenario are shown in boldface.**

| | | Sample Size | POD | FAR | HSS | ETS |
|---|---|---|---|---|---|---|
| Ocean Day | MYD06 | 482527 | 94.18% | 6.28% | **0.822** | **0.697** |
| | NJIAS | 482527 | 88.34% | 5.89% | 0.755 | 0.606 |
| | JAXA | 482527 | 96.48% | 15.86% | 0.658 | 0.490 |
| Ocean Night | MYD06 | 451539 | 92.03% | 8.04% | **0.721** | **0.563** |
| | NJIAS | 451539 | 88.18% | 5.39% | **0.721** | **0.563** |
| Land Day | MYD06 | 128990 | 93.12% | 8.10% | **0.772** | **0.629** |
| | NJIAS | 128990 | 89.19% | 6.06% | 0.758 | 0.610 |
| | JAXA | 128990 | 95.30% | 12.81% | 0.706 | 0.546 |
| Land Night | MYD06 | 158640 | 94.33% | 13.81% | 0.729 | 0.574 |
| | NJIAS | 158640 | 85.05% | 5.66% | **0.752** | **0.602** |

**3.2 Cloud height results**

The cloud height retrievals are evaluated against the CALIOP 1-km cloud layer products. The CALIOP CTH is interpreted as the top altitude of the uppermost CALIOP cloud layer. The CALIOP CTP

and CTT are from the *Modern Era-Retrospective analysis for Research and Applications*, Version 2 (MERRA-2) and are interpolated to the CALIOP CTH altitude (Avery et al., 2020). Figure 6 shows the joint probability histograms of three cloud height parameters (CTT, CTH and CTP) between CALIOP and MYD06 and between CALIOP and NJIAS datasets in 2017. To facilitate comparisons, CTH is

expressed in kilometers above sea level. Overall, the NJIAS cloud height retrieval algorithm outperforms its MYD06 counterpart. The correlation coefficients (CCs) of CTH, CTP and CTT between NJIAS and CALIOP products are 0.84, 0.84 and 0.80, respectively—each surpassing the corresponding values obtained from MYD06 retrievals. It is noteworthy that the NJIAS retrievals tend to slightly underestimate CTH and overestimate both CTP and CTT for high clouds, possibly due to the fact that only a single channel centered at 13.3 μm is allocated within the broad carbon dioxide absorption region for the AHI. Consequently, the multiplicative biases (MBs; Zhuge et al., 2021b) associated with these three cloud height parameters stand at 1.16, 0.91, and 0.97, respectively. Incorporating additional carbon dioxide absorption channels would enhance the inference of cloud-top pressure and effective cloud amount for high-level clouds, especially semi-transparent clouds such as cirrus (Platnick et al., 2019). The MYD06 algorithm also comes with its limitations. There exists a significant proportion of instances in which the MYD06 algorithm mistakes mid- and high-level clouds for boundary layer clouds. The root-mean-square errors (RMSEs) for MYD06 CTH, CTP and CTT retrievals are 3.51 km, 196.80 hPa and 22.89 K, respectively, substantially larger than those reported for the NJIAS retrievals.

The JAXA operational cloud height algorithm incorporates the IR window technique, the radiance rationing technique, and the IR-water vapor intercept technique, and choose one of them contingent upon the result of cloud type classifications (Mouri et al., 2016b). This conventional methodology is different from the maximum likelihood estimation algorithms, such as the ACHA. The JAXA dataset includes two cloud height parameters, CTH and CTT, which are available only in daytime. By comparing NJIAS daytime CTH and CTT retrievals to JAXA's results, Figure 7 confirms the remarkable improvement in the accuracy of these two cloud height parameters yielded by the NJIAS. The JAXA retrievals exhibit a more obvious tendency to underestimate the CTH and overestimate the CTT of mid-to-high-level clouds than the NJIAS retrievals. Meanwhile, there is a poor agreement between CALIOP and JAXA CTH retrievals for low-level clouds, with most samples straying away from the one-to-one ratio lines. As a result, the RMSE values for the JAXA CTH and CTT retrievals are 3.17 km and 22.42 K, respectively, which are much larger than the metrics of 2.65 km and 17.90 K for the NJIAS retrievals.

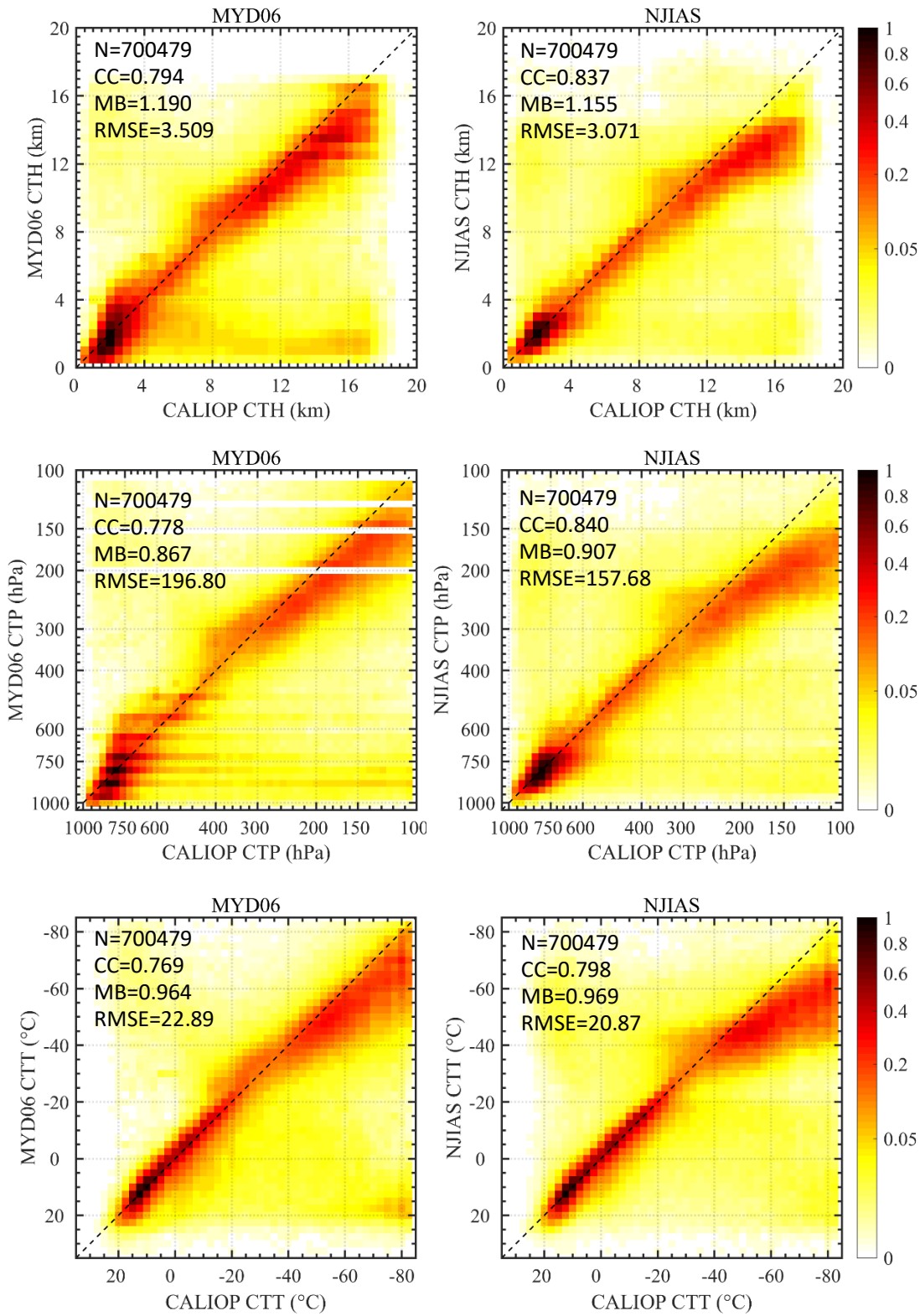

**Figure 6: Joint probability density histograms of CTH (km; top panels), CTP (hPa; middle panels) and CTT (°C; bottom panels) between CALIOP and MYD06 (left panels) and between CALIOP and NJIAS (right panels) datasets in 2017. Also indicated in each panel are sample size (N), correlation coefficient (CC), multiplicative bias (MB) and root-mean-square error (RMSE). Clear pixels identified by either MYD06 or**

**NJIAS are excluded from the statistics.**

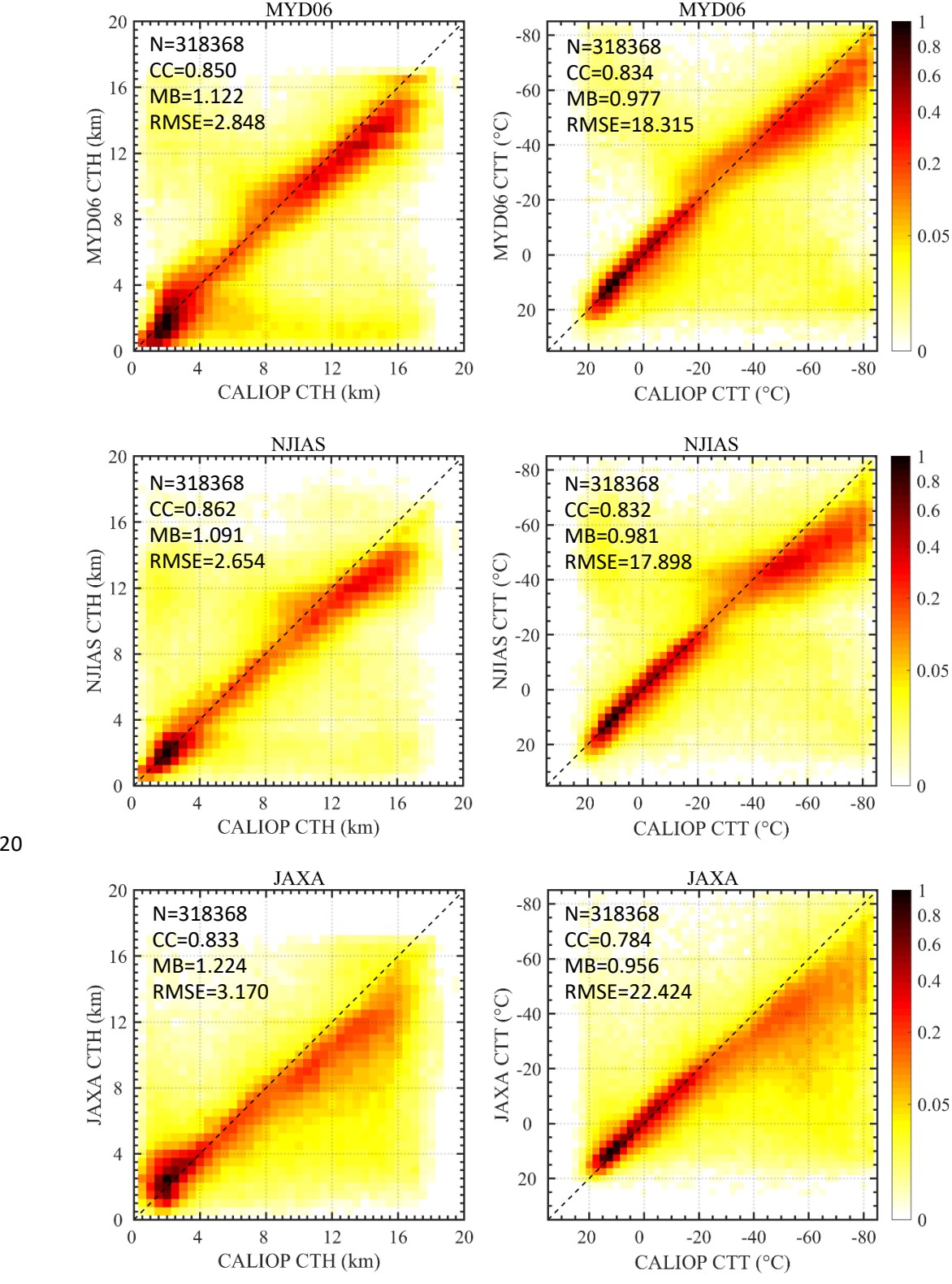

**Figure 7: Joint probability density histograms of CTH (km; left panels) and CTT (°C; right panels) between CALIOP and MYD06 (top panels), between CALIOP and NJIAS (middle panels), and between CALIOP and JAXA (bottom panels) datasets in daytimes of 2017. Clear pixels identified by MYD06, NJIAS, or JAXA are excluded from the statistics. Only daytime data are retained.**

**3.3 Cloud-top phase results**

The CALIOP cloud-top phase is defined as the CALIOP cloud phase of the uppermost cloud layer, which will serve as the truth in the following evaluations. The CALIOP classification currently provide
four categories of phases, that is, liquid-water, randomly oriented-ice (ROI), horizontally oriented-ice, and unknown (Hu et al., 2009). The latter two categories are not considered in this study because of their low percentages of occurrence (less than 1.0%) (Zhuge et al., 2021a). In addition, the Collection-6 MYD06 dataset provides two independent cloud-top phase retrievals. One is an IR-only results available all day, and the other is derived from a combination of SWIR and IR tests that runs during daytime only
(Baum et al., 2012).

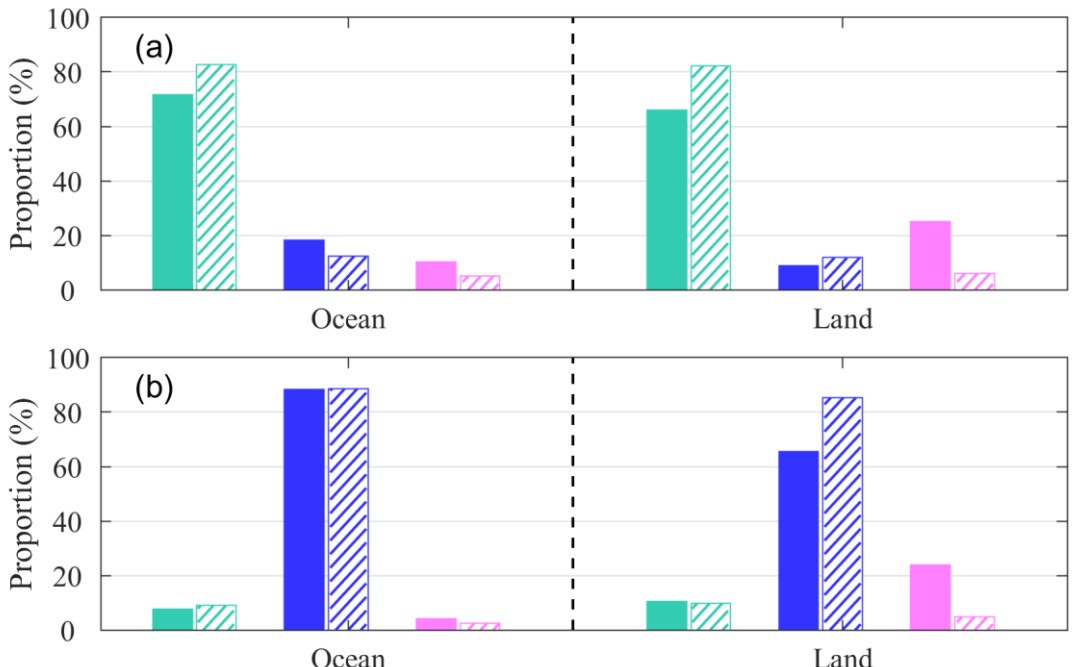

**Figure 8: Proportions of liquid-water (turquoise), ice (blue), and mixed/uncertain (magenta) phases identified by MYD06 IR-only (solid bars) and NJIAS (hatched bars) for CALIOP pixels with (a) liquid-water- and (b)**
**ROI-phase cloud tops in 2017 over oceans and land. Clear pixels identified by either MYD06 or NJIAS are excluded from the statistics.**

Figure 8 demonstrates that the NJIAS cloud-top phase retrievals perform better than the MYD06 IR-only retrievals. For CALIOP liquid-water and ROI cloud tops over oceans, the PODs of NJIAS

retrievals are 82.60% and 88.59%, respectively. These two metrics slightly decrease to 82.17% and 85.35%

over land. Over oceans, the MYD06 IR-only and NJIAS datasets exhibit similar behavior for CALIOP

ROI cloud-top phases. However, compared to NJIAS HCFD, the MYD06 IR-only dataset tends to

classify more CALIOP liquid-water phases as ice or uncertain phases, resulting in a POD of 71.59%.

Over land, the MYD06 IR-only dataset classifies many CALIOP cloud tops as having an uncertain phase,

resulting in low PODs of only 66.03% and 65.63% for CALIOP liquid-water and ROI cloud tops,

respectively.

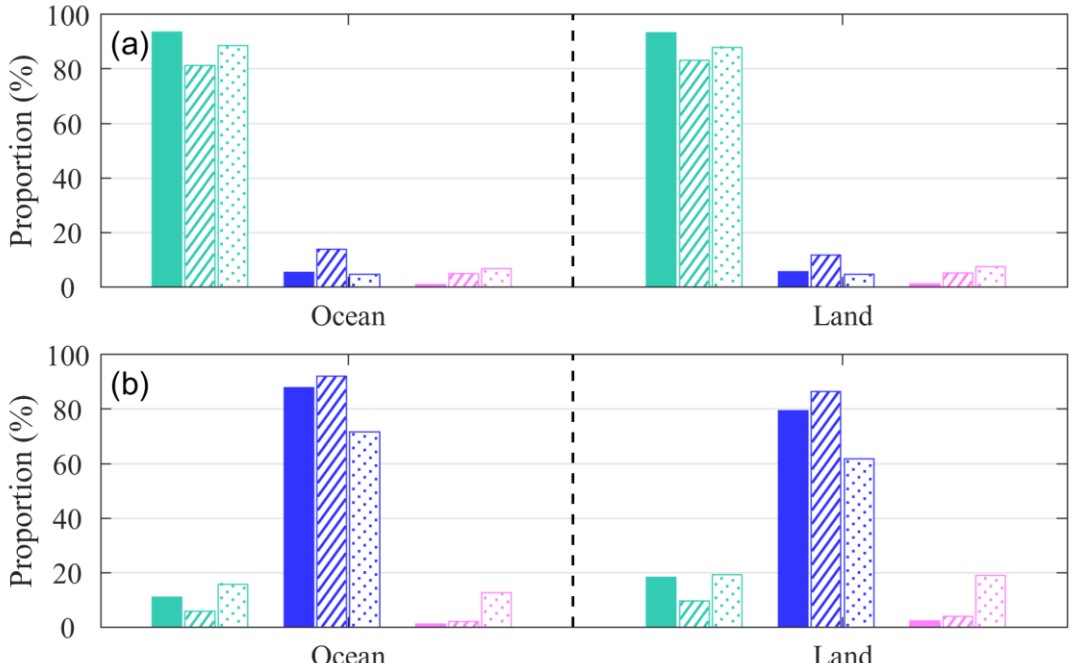

**Figure 9: Proportions of liquid-water (turquoise), ice (blue), and mixed/uncertain (magenta) phases identified**

**by MYD06 SWIR+IR (solid bars), NJIAS (hatched bars) and JAXA (dotted bars) for CALIOP pixels with (a)**

**liquid-water- and (b) ROI-phase cloud tops in daytimes of 2017 over oceans and land. Clear pixels identified**

**by MYD06, NJIAS, or JAXA are excluded from the statistics. Only daytime data are retained.**

Intercomparisons of cloud-top phase retrievals are also made among the MYD06 SWIR+IR, the

NJIAS, and the JAXA datasets during daytime only (Fig. 9). It can be seen that NJIAS cloud-top phase

retrievals exhibit a consistent accuracy for both day and night. Meanwhile, the MYD06 SWIR+IR

retrievals (Fig. 9) show a significant improvement over the IR-only retrievals (Fig. 8) by supplementing

the IR tests with those from solar channels. Figure 9 also reveals a deficiency of the JAXA retrievals in

identifying ice phases. The PODs of the JAXA dataset for CALIOP ROI phases are as low as 71.69%

over oceans and 61.84% over land, which are significantly worse than those for CALIOP liquid-water

phases.

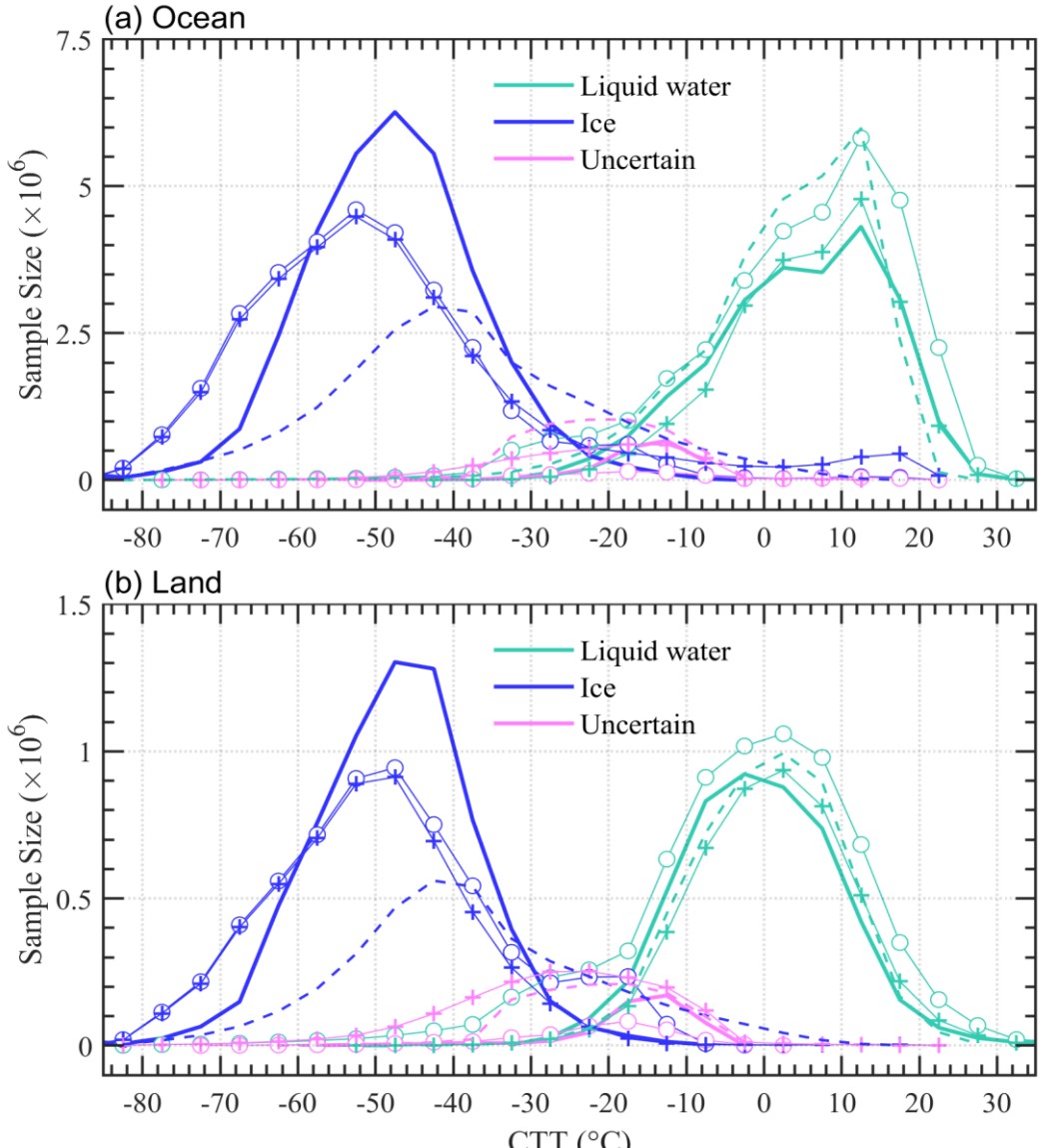

**Figure 10: Sample size variations of cloud-top phases identified by MYD06 IR-only (plus signs connected by**

**thin lines), MYD06 SWIR+IR (open circles connected by thin lines), NJIAS (thick solid curves) and JAXA**

**(dashed curves) with respect to the CTT values in daytimes of June and December 2017 over (a) oceans and**

**(b) land.**

It is worthwhile to examine the distributions of the MYD06 IR-only, MYD06 SWIR+IR, NJIAS

and JAXA identified cloud-top phases with respect to the CTT values (Fig. 10). The NJIAS HCFD tends

to classify cloudy pixels with CTT above 0 ℃ as liquid-water and those with CTT below -30 ℃ as ice.

When CTT is between -30 ℃ and 0 ℃, the NJIAS-identified cloud-top phase could be liquid water, ice, or a mixture of both. However, there are cases where the MYD06 IR-only or the JAXA classified cloud tops with a CTT greater than 0 ℃ as ice phase, revealing a limitation of these two products. Continent cloud tops with uncertain (liquid-water) phase are also found in the MYD06 IR-only (SWIR+IR) retrievals when CTT is below −40 ℃. Considering that in situ observations have not revealed the presence of a mixed or supercooled-water phase at temperatures below −40°C (Korolev et al., 2017), it is necessary to reexamine the two MYD06 cloud-top phase classifications over land.

### 3.4 DCOMP results

The NJIAS DCOMP retrievals are evaluated using the Collection-6.1 MYD06 products in June, July and August 2017. Note that both the NJIAS and the MYD06 have three $\tau$ retrievals. In most cases these three $\tau$ retrievals are nearly identical. Accordingly, the DCOMP35 $\tau$ is selected as a representative in this study. Besides, since all current bispectral-based DCOMP algorithms have large uncertainties or errors in the Re retrievals of thin clouds, samples with $\tau$ less than 5 are removed during the Re valuations.

Figure 11 illustrates pixel-to-pixel comparisons of Re and $\tau$ between the MYD06 and NJIAS retrievals. The NJIAS $Re_{1.6}$ retrievals are generally consistent with the MYD06 $Re_{1.6}$ values for both liquid-water and ice clouds. Most samples are distributed evenly around the one-to-one ratio lines. The CC of the NJIAS $Re_{1.6}$ retrievals for liquid-water (ice) clouds is 0.72 (0.85), and the corresponding MB and RMSE values are 1.06 (0.95) and 3.42μm (6.10μm), respectively. The NJIAS $Re_{3.9}$ retrievals for liquid water clouds are systematically smaller than their MYD06 counterparts that has an MB of 0.85 and a CC of 0.85. However, such an underestimation is not found in the NJIAS $Re_{3.9}$ retrievals for ice clouds, which yielded an MB of 1.00, a CC of 0.76 and a RMSE of 6.04 μm. Overall, the NJIAS $\tau$ retrievals agree well with the MYD06 $\tau$ values for both liquid-water and ice clouds. The MB ranges from 1.08 to 1.12, and the CC ranges from 0.73 to 0.76.

The JAXA dataset only provides one pair of Re and $\tau$ derived using 0.64-μm and 2.3-μm channels. Figure 12 compares the results between the NJIAS and JAXA retrievals. Note that the sample sizes are less than those in Fig. 11 due to a large amount of retrieval failures in the JAXA algorithm. The NJIAS $Re_{2.3}$ retrievals in both liquid-water and ice clouds show a systematic overestimation (~2 μm) when MYD06 $Re_{2.1}$ retrievals are regarded as the "truth". The overestimations are likely due to a discrepancy in the sensor central wavelengths which will affect the reflectance observations and the DCOMP LUTs

(Wang et al., 2018). Interestingly, the overestimations are not found in the JAXA retrievals. A detailed comparison of the LUTs used by the NJIAS and the JAXA is essential. The performances of $\tau$ retrievals from NJIAS and JAXA are similar in general, except for a slight overestimation of ice clouds in the JAXA products.

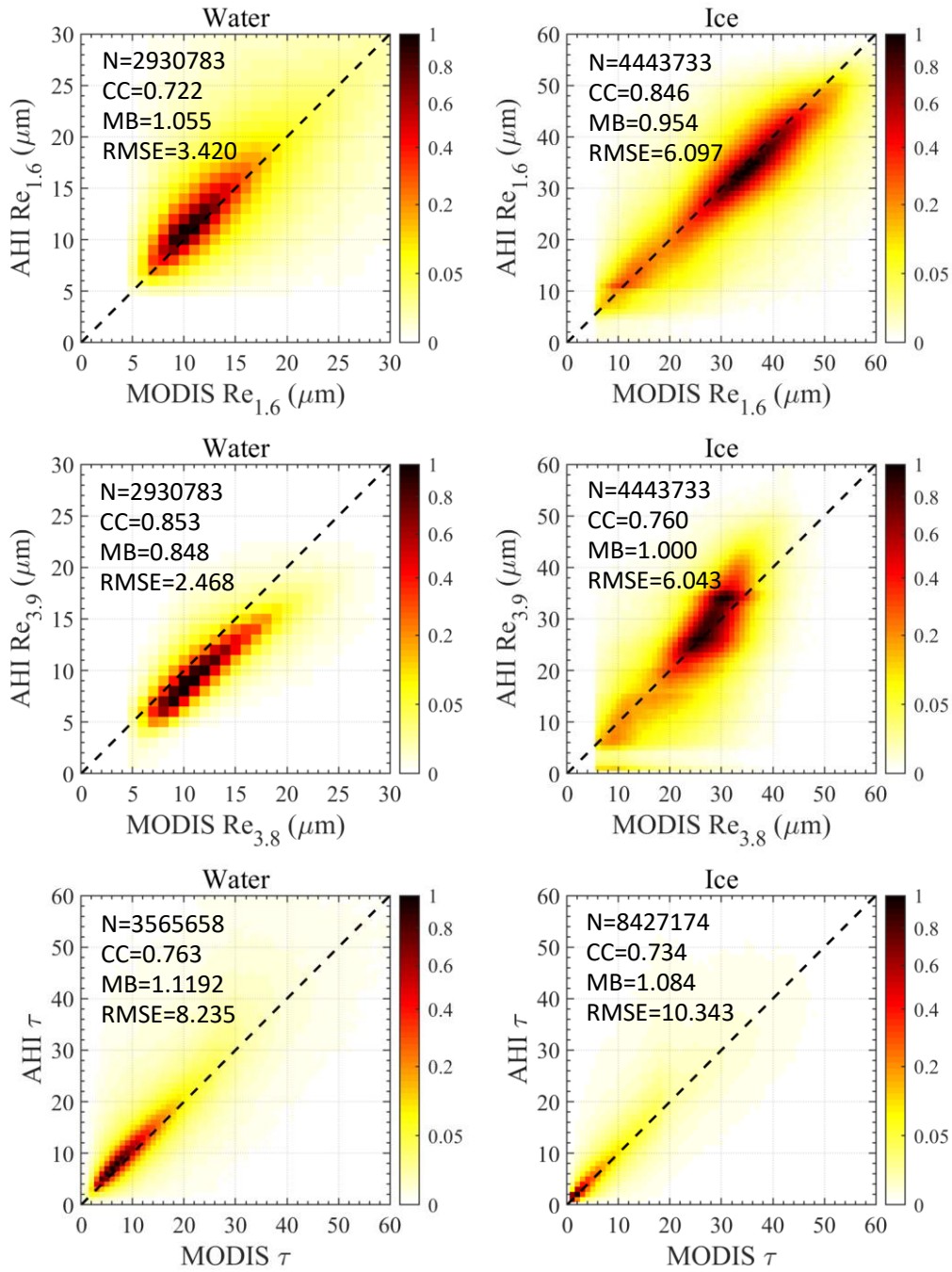

**Figure 11: Joint probability density histograms of $Re_{1.6}$ (top panels), $Re_{3.8}$ [$Re_{3.9}$] (middle panels), and $\tau$**

**(bottom panels) between MYD06 and NJIAS datasets for liquid water (left panels) and ice (right panels) clouds in daytimes of June, July and August 2017.**

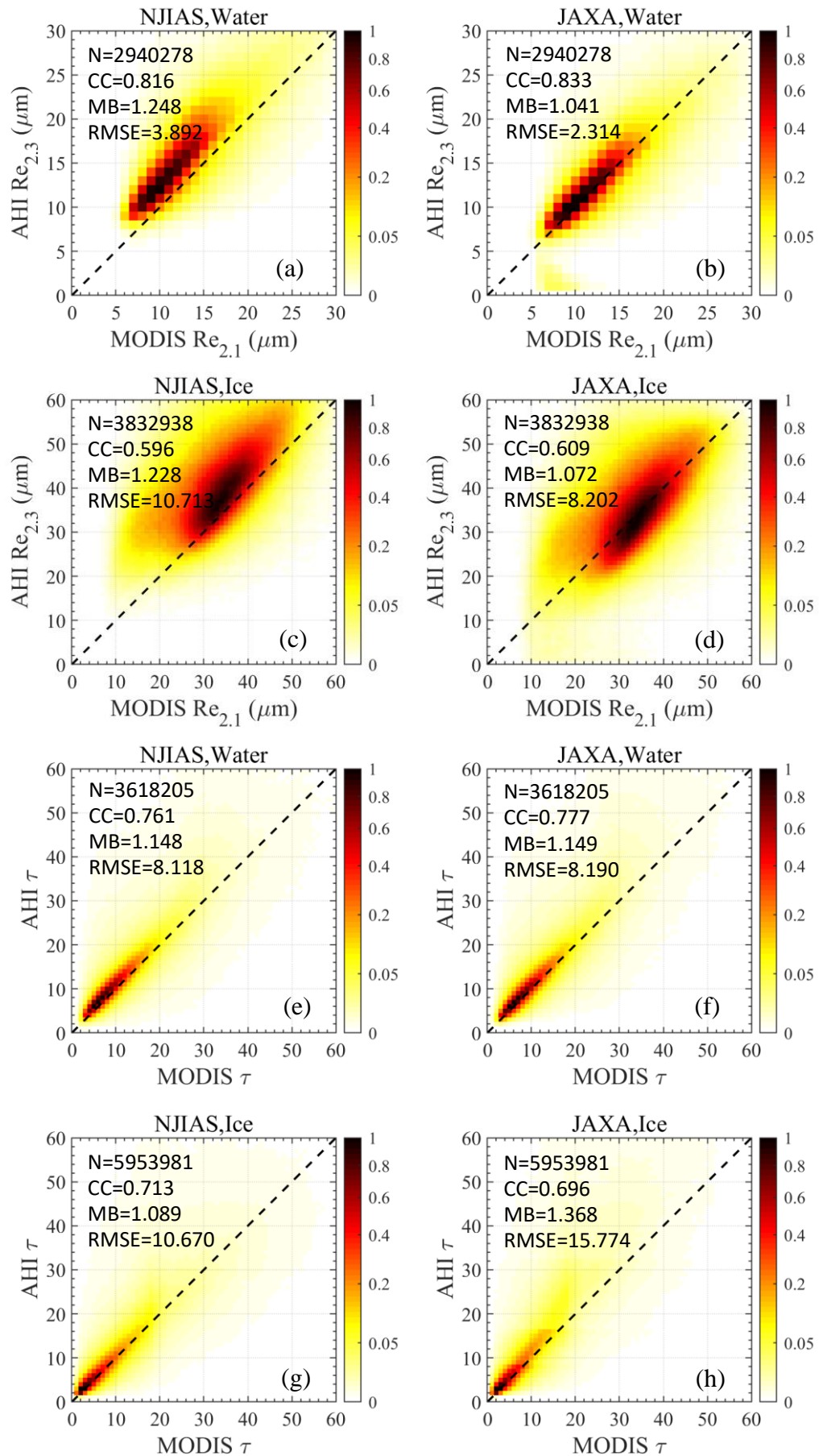

**Figure 12: Joint probability density histograms of (a–d) Re$_{2.1}$ [Re$_{2.3}$] and (e–h) $\tau$ between MYD06 and NJIAS (left panels) and between MYD06 and JAXA (right panels) datasets for (a–b, e–f) liquid-water and (c–d, g–h) ice clouds in daytimes of June, July and August 2017.**

## 3.5 Case study

To better illustrate the differences in cloud retrievals among three datasets, a case occurring over the WNP at 04:50 UTC on 7 June 2017 is presented (Fig. 13). At this time, the lower-right portions of the AHI VIS and SWIR images were contaminated by the sun glint (Fig. 13a).

Cloud mask results in the three datasets exhibit significant discrepancies in region "A", where MYD06 indicates cloudiness (Figs. 13c) while NJIAS and JAXA indicate clear conditions (Figs. 13d and 13e). It can been inferred that the MYD06 identifies region "A" as cirrus because the cloud top phase derived by the MYD06 was ice (Fig. 13f). Besides, the JAXA product classifies some clear-sky pixels and a majority of cloudy pixels as probably cloudy over the sun-glint areas (Fig. 13e). This is the reason for JAXA dataset to have high PODs but also high FARs.

The MYD06 misclassifies water clouds in region "B" (which appear white on the false-color image) as ice clouds. However, both the MYD06 and NJIAS products demonstrate good performances in multilayer cloud cases. Both datasets report an ice phase in region "C" where thin cirrus clouds were overlying low-level water clouds (Figs. 13f and 13g). In contrast, the JAXA product gives a liquid-water phase in region "C" (Fig. 13h), suggesting that the JAXA cloud-top phase algorithm requires further enhancement.

The NJIAS dataset underestimates the CTH of high-level clouds by 0.5–1 km when compared to the MYD06 product. Nevertheless, the MYD06 has obvious limitations in the CTH estimations for thin cirrus. For example, the ice-phase clouds (i.e., cirrus) in region "A" have a CTH of less than 1 km, which is not reasonable. The JAXA dataset fails in the CTH retrievals over the sun-glint areas. According to the CALIOP observations (Fig. 13b), region "D" was covered by fogs, with a CTH of less than 1 km. However, the JAXA CTH values in region "D" are ~3km, higher than those reported by both MYD06 and NJIAS. JAXA also tends to underestimate the CTH of multilayer clouds by ~5 km. All of the above reveal some shortcomings of the JAXA CTH algorithm.

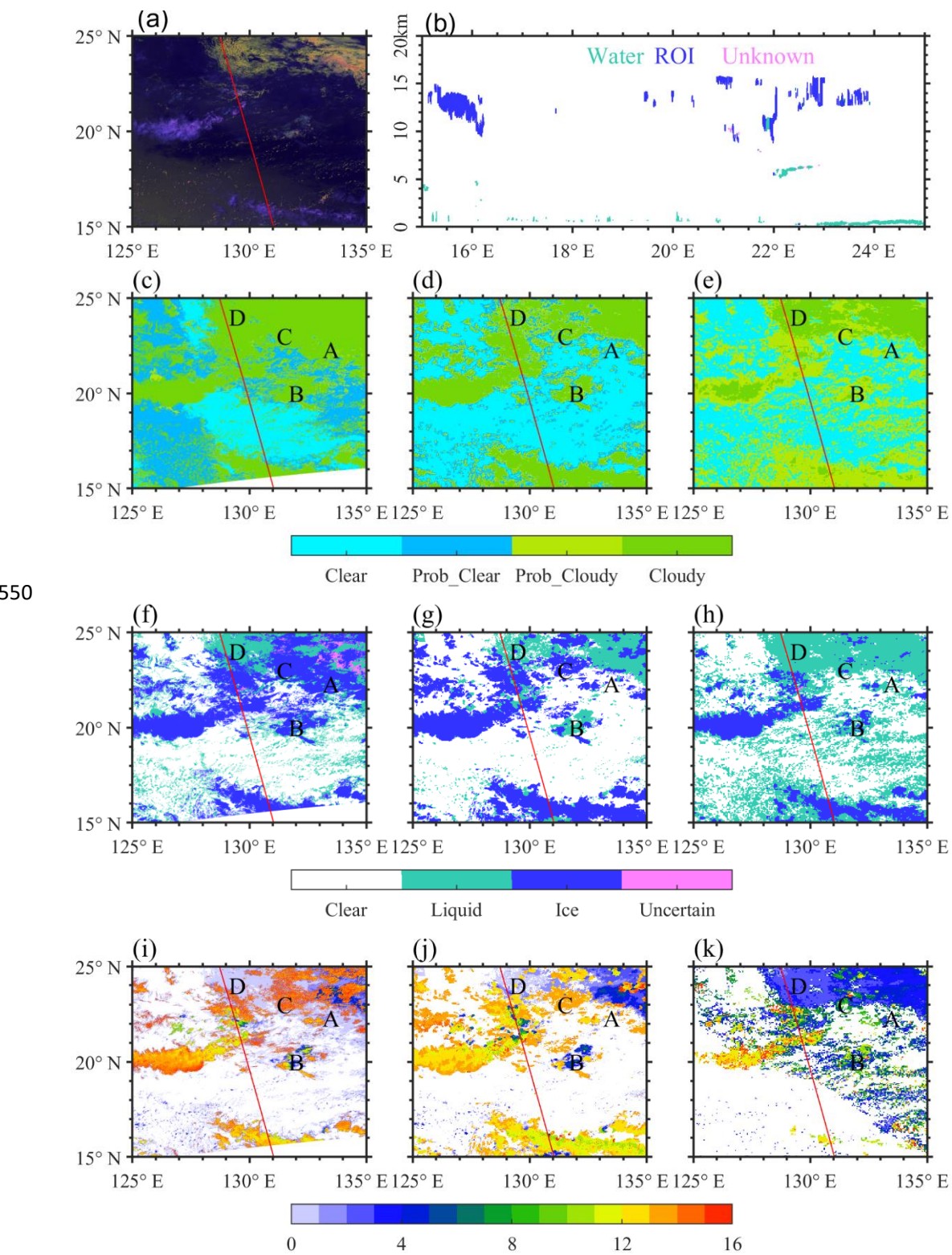

**Figure 13: A case at 04:50 UTC on 7 June 2017, illustrating the differences in cloud retrievals among three datasets. (a) AHI false-color image (red, 0.64 μm; green, 1.6 μm; blue, 11.2 μm reversed), (b) CALIOP cloud phase profile, as well as (c–e) cloud mask, (f–h) cloud-top phase, and (i–k) CTH (unit: km AGL) results from the MYD06 (left panels), NJIAS (center panels) and JAXA (right panels). The red line in (a) and (c–k) indicates the CALIOP track.**

**4 Application Examples**

**4.1 Cloud climatology in southwestern China**

The climate in southwestern China is controlled by the East Asian and South Asian monsoons, in combination with the complex terrain. During the cold season (November–April), a quasi-stationary front frequently occurs over the Yunnan–Guizhou Plateau (Cai et al., 2022), resulting in a sharp contrast of weather conditions on its two sides: cloudy or rainy sky in Guizhou province (103 °–109 °E, 24 °–29 °N) but clear sky in Yunnan province (97 °–106 °E, 21 °–29 °N). Meanwhile, the moist environment and calm

winds provide favorable conditions for the frequent foggy weather over the Sichuan Basin (103 °–108 ° E, 28 °–32 °N).

Figure 14 presents a simple analysis of the cloud climatology over southwestern China based on the cloud products in the cold seasons of years 2016–2020. Three daytime variables including cloud mask, CTH and $\tau$ are employed. The MODIS/Aqua provides daytime observations at most once per day, at

~13:30 local solar time. Therefore, results from the MYD06 are for reference only. It can be seen that the NJIAS HCFD provides a reasonable description of the spatial distribution of cloud covers over southwestern China in the cold season. The cloud occurrence frequencies are ~30% over Yunnan and ~80% over Guizhou. However, the JAXA dataset presents a weaker contrast of cloud occurrence frequencies on the two sides of the quasi-stationary front. The cloud occurrence frequencies are as high

as ~50% over Yunnan, which is only 30% less than those over Guizhou. Moreover, the JAXA returns a factitious high-frequency of greater than 90% of cloud occurrences in the eastern part of the Tibetan Plateau (95 °–103 °E, 26 °–35 °N), which is likely a result from mislabeling glacier or snow-covered areas as clouds (figures omitted). The spatial distributions of averaged CTH also exhibit large differences between the NJIAS and JAXA datasets. The JAXA tends to underestimate the CTH, especially in the

areas where cloud covers are obviously overestimated. For the spatial pattern of the averaged $\tau$, there is a distinct regional difference between the eastern and western parts of southwestern China. Thick clouds often occur over the eastern part of southwestern China while thin clouds often occur over the western part, which are revealed by both the NJIAS and JAXA datasets. Nonetheless, the thick (thin) clouds tend to have a greater (smaller) $\tau$ in the JAXA dataset than those in NJIAS dataset.


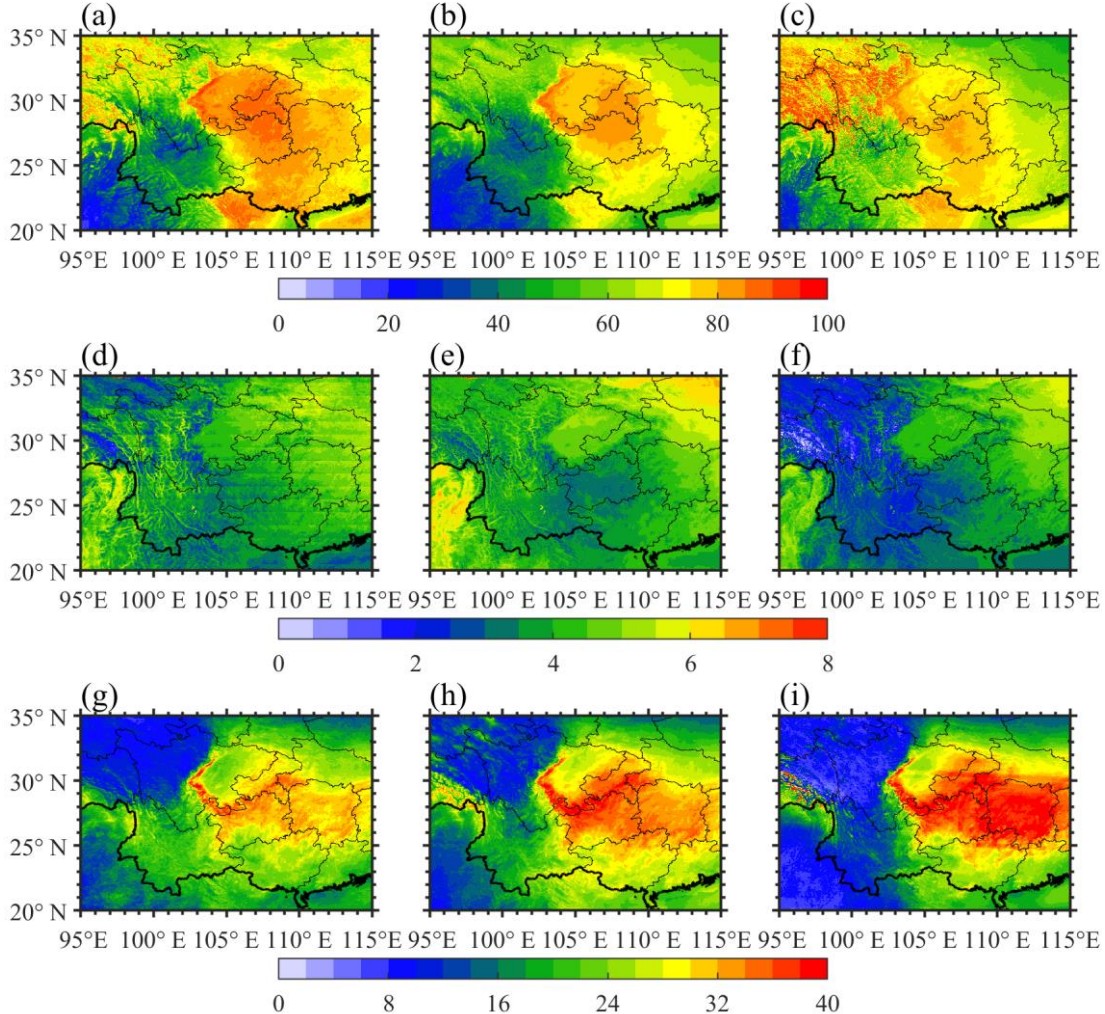

**Figure 14: Spatial distributions of (a–c) cloud occurrence frequency (unit: %), (d–f) averaged CTH (unit: km AGL) and (g–i) τ (unitless) within 0.05°×0.05° grid boxes over southwestern China using 5-yr boreal cold-season cloud products of MYD06 (left panels), NJIAS (center panels), and JAXA (right panels). Only daytime data are retained.**

## 4.2 Cloud and precipitation features of landfalling typhoons

The NJIAS HCFD–TyWNP provides a comprehensive description of cloud macro- and micro-physical characteristics within a $20° \times 20°$ longitude-latitude grid box surrounding the center of WNP typhoons. This product is useful for understanding cloud and precipitation features of typhoons. Figure 15 illustrates the utility of NJIAS HCFD–TyWNP for analyzing the intensity of typhoon rainfall in In-Fa (2021) and Hagupit (2020). The typhoon In-Fa (202106) brought record-breaking hourly rainfall to Henan Province on 21 July 2021 when it was still positioned offshore (Wei et al., 2023). In-Fa made its first landfall at 04:30 UTC on 25 July on Zhoushan Islands at the northern coast of Zhejiang Province,

with a minimum central pressure of 970 hPa according to the best-track records (Lu et al., 2021). Prior to its first landfall in Zhejiang, the central dense overcast (CDO) of In-Fa gradually disintegrated and the convection weakened. The eastern half of CDO was characterized by extensive cumulonimbus clouds

with a CTH of 14 km. Due to land effects, the western half of CDO was dominated by liquid-water clouds, with a significantly low CTH and very weak vertical motion. As a result, within 24 hours before and after In-Fa made the first landfall, most areas of Zhejiang province experienced a stable stratiform precipitation. The rain rates measured by rain gauges were generally weak, mainly 5–20 mm h$^{-1}$, and the local maximum rain rate was only 49.0 mm h$^{-1}$. The rain rate at the landing site was only 29 mm h$^{-1}$. In

contrast, typhoon Hagupit (202004) made its landfall at 19:30 UTC on 3 August 2020 in southeastern Zhejiang, with a minimum central pressure of 965 hPa, similar to the intensity of In-Fa (202106) making landfall. However, during the landfall of Hagupit, the CDO distribution was complete and compact. As a result, rainstorms were produced along the track of Hagupit. The maximum rain rate measured by rain gauges in Zhejiang during the 24 hours before and after Hagupit's landfalling time was 98.7 mm h$^{-1}$.


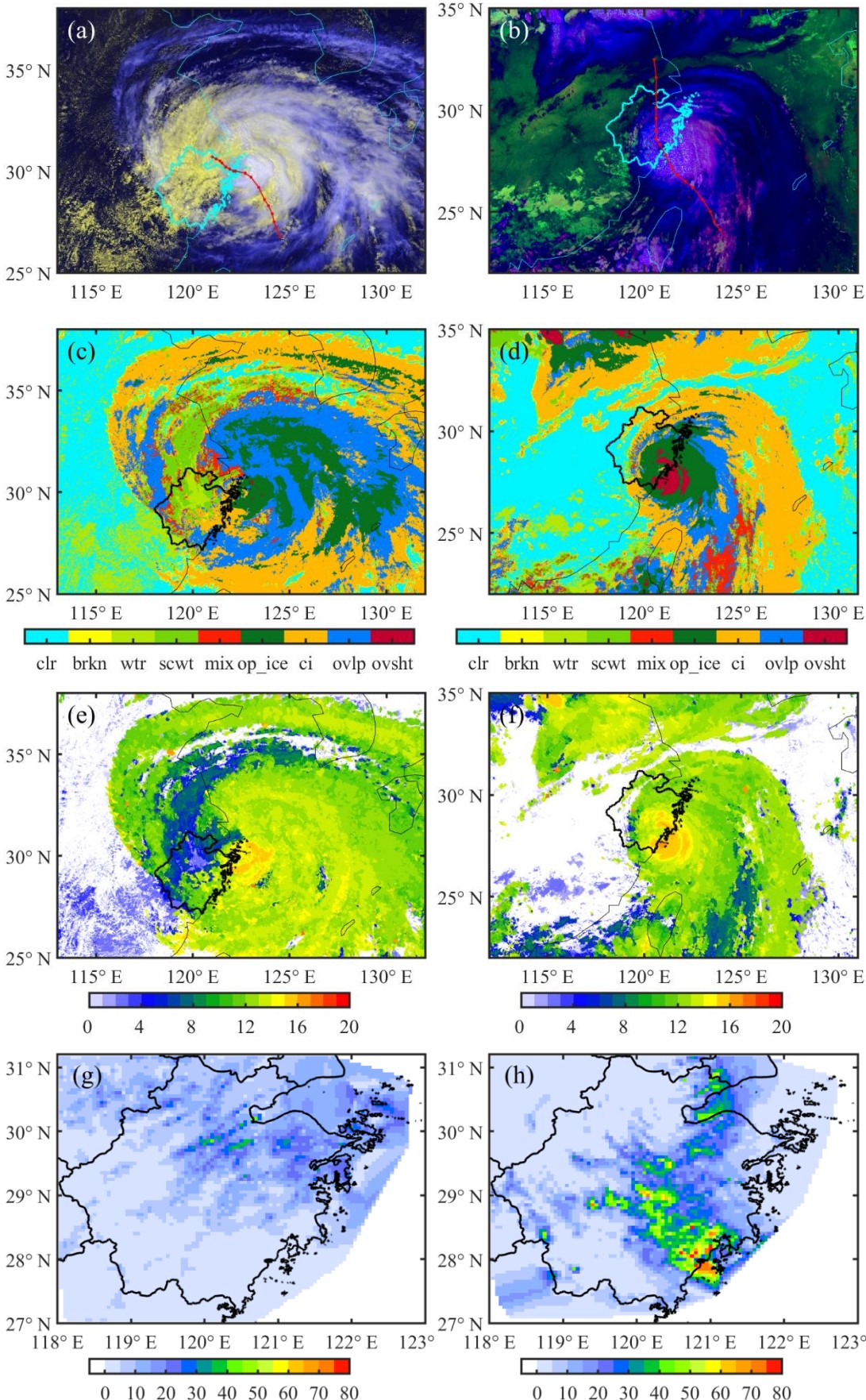


**Figure 15: (a–b) AHI TC-RGB composite images, as outlined in Chen et al. (2022), featuring two modes with distinct color representations: (a) for the day mode (red, 0.64 μm; green, 0.64 μm; blue, 11.2 μm reversed), cirrus appears blue, convective clouds appear white, and low clouds appear yellow; and (b) for the night mode (red, 12.3 μm -10.4 μm; green, 10.4 μm - 3.9 μm; blue, 11.2 μm reversed), cirrus appears blue, low clouds appear bright green, and convective clouds appear dark violet, (c–d) cloud types including clear (clr), broken (brkn), warm-water (wtr), supercooled-water (scwt), mixed (mix), opaque-ice (op_ice), cirrus (ci), overlapped (ovlp), and overshooting (ovsht), and (e–f) CTH (unit: km AGL) at the landfalling time $t_{lf}$, as well as (g–h) maximum gauge rain rate within the $t_{lf}$ ±24 h time window (unit: mm h$^{-1}$) for Typhoons In-Fa (202106) (left panels) and Hagupit (202004) (right panels). The thick lines denote the boundaries of Zhejiang province. The red curve denotes the typhoon track at 3-h interval during the $t_{lf}$ ±24 h time window.**

## 5 Data availability

The NJIAS HCFD described in this article was released to the general public. Since the Science Data Bank accepts up to 1 TB per data publication, the NJIAS HCFD–0.04Deg was divided into four parts and published at https://doi.org/10.57760/sciencedb.09950 (Zhuge, 2023a), https://doi.org/10.57760/sciencedb.09953 (Zhuge, 2023b), https://doi.org/10.57760/sciencedb.09954 (Zhuge, 2023c), and https://doi.org/10.57760/sciencedb.10158 (Zhuge, 2023d). The NJIAS HCFD–TyWNP is published at https://doi.org/10.57760/sciencedb.09945 (Zhuge, 2023e).

## 6 Summary and conclusions

To supplement the JAXA Himawari-8/9 operational cloud products, which are daytime only, a dataset named the NJIAS HCFD was constructed. The NJIAS HCFD provides 30 variables (e.g., cloud mask, cloud-top phase, CTH, τ and Re, as well as snow, dust and haze masks) and covers a vast majority of the East Asia and WNP regions over the 7 yr period from April 2016 to December 2022. In this study, the NJIAS HCFD data quality has been evaluated against the CALIOP 1-km cloud layer product and the Collection-6.1 MYD06 dataset. The evaluation results are summarized as follows.

1) The POD and FAR of the NJIAS HCFD for cloud detections are ~88% and ~6%, respectively. The NJIAS HCFD gives higher skill scores than the MYD06 during nighttime. For daytime scenario, the NJIAS HCFD lags behind the MYD06, but outperforms JAXA dataset. Note that in the statistical analysis,

CALIOP cases with sub-pixel cloudiness or very thin cirrus (Karlsson et al., 2018; 2023) have been excluded.

2) The three cloud height parameters (CTT, CTH and CTP) derived from the NJIAS HCFD show better agreement with the CALIOP data than those obtained from the MYD06. The NJIAS retrievals tend to slightly underestimate CTH and overestimate both CTP and CTT for high clouds. The JAXA product has a more pronounced tendency to underestimate the CTH and overestimate the CTT of mid-to-high-level clouds.

3) The PODs of the NJIAS phase determinations for the CALIOP liquid-water and ROI cloud tops are 82.60% (82.17%) and 88.59% (85.35) over oceans (land), respectively. Problems are found for the MYD06 and JAXA retrievals, such as misclassifying pixels with a CTT greater than 0 ℃ as ice phase over ocean, and misclassifying pixels with a CTT below -40 ℃ as non-ice phase over land.

4) Overall, the NJIAS DCOMP retrievals have high correlations with the Collection-6.1 MYD06 results, with CC ranging from 0.722 to 0.853. The JAXA dataset only provides Re values retrieved from
the AHI 2.3-μm channel. However, the overestimation in the NJIAS $Re_{2.3}$ retrieval is not found in the JAXA retrievals.

The NJIAS HCFD is subject to uncertainties. For example, the NCEP FNL analysis with a 6-h temporal resolution, although having been interpolated to align with AHI observation times, are
insufficient for capturing the rapid changes in land surface temperatures observed in certain regions and during specific times of the day, such as early morning hours. The accuracy of the fog and snow masks, which heavily depend on land surface temperature observations, could be compromised due to an inability to imprecisely represent diurnal temperature variations. Furthermore, given the systematic overestimation found in the NJIAS $Re_{2.3}$ retrieval, an in-depth inter-sensor radiometric analysis is crucial.
A radiometric adjustment factor, which excludes the effect of central wavelength shift, can be employed for aligning AHI's relative radiometric calibration more closely with that of the MODIS. The quantitative assessment of the uncertainties associated with the NJIAS HCFD will be the focus of future investigations.

Despite the issues addressed above, it is anticipated that the NJIAS HCFD will play an important
role in monitoring the evolutions of convection and weather systems, studying aerosol-cloud-precipitation-climate interactions, and evaluating cloud parameterization schemes in weather/climate models. Two examples presented in this article demonstrate the use of the NJIAS HCFD for climate and

typhoon research. In the future, the time period of the dataset will be extended continuously. More cloud variables, such as cloud-base height and nighttime optical/microphysical parameters, may be added to

the dataset by using the deep-learning-based cloud retrieval algorithms recently developed by Wang et al. (2022, 2023).

**Author contributions.** XZ (Xiaoyong Zhuge) conceived the idea and prepared the data. XZ (Xiaoyong

Zhuge), XZ (Xiaolei Zou) and LY drafted the manuscript. All authors contributed to manuscript revisions.

**Competing interests.** The authors declare that they have no conflict of interests.

**Acknowledgements.** JAXA distributes the Himawari-8/9 raw data and level-2 cloud products

(https://www.eorc.jaxa.jp/ptree/). NASA's official website (https://earthdata.nasa.gov/) provides the MYD06 and CALIOP cloud products. The authors also thank the editors and anonymous reviewers for their helpful comments and valuable suggestions, which improved the manuscript.

**Financial support.** This work was financially supported by the National Natural Science Foundation of

China (42175006), Jiangsu Youth Talent Promotion Project (2021-084), CMA Scientific and Technological Innovation Team Construction Project (CMA2023ZD06), the Fengyun Application Pioneering Project (FY-APP-2021.0101), and the Basic Research Fund of CAMS (2020R002, 2021Z002, 2021Y013, 2021Y014).

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
