# Peer review of "Introduction to the NJIAS Himawari-8/9 cloud feature dataset for climate and typhoon research"

_Earth System Science Data, 2023_

## Referee Comment (RC1)

Review of Earth System Science Datasets manuscript by Zhuge et al. entitled

**" Introduction to the NJIAS Himawari-8/9 cloud feature dataset for climate and typhoon research"**

**General impression and recommendations**

This manuscript describes a new cloud climatology (NJIAS) based on data from the AHI imager onboard the Himawari 8+9 geostationary satellites. The covered region is the East Asia and North Pacific part of the Himawari disc and the temporal period covered is from 2016 to 2022. The methods for deriving various cloud parameters are described in detail. Validation results based on comparisons with data from the cloud lidar CALIOP on the CALIPSO satellite, with corresponding products from MODIS and with and operational AHI-based products from JAXA (Japan) are presented. The climatology is found to produce results compatible or, for some parameters, even better than the reference datasets from MODIS and Jaxa.

I find the study interesting because of the comprehensive use of existing methods (thus, making use of long-term international experience) but also the addition of new features which improve results even further. The manuscript is well-written, figures and tables are very appropriate and well designed, and the English language is actually surprisingly good for coming from non-native authors.

I do recommend acceptance of this manuscript but I also have a number of questions, mainly related to the methods used for validating the results. I have identified some problems with the methods and also some weaknesses in the validation methods and I think it is appropriate that authors at least discuss these weaknesses in an updated manuscript before the paper is published.

Detailed remarks and questions follow below and at the end some editorial remarks.

**General remarks and questions:**

**Page 4, line 112:** Is the temporal resolution of 6 hours from NCEP reanalyses really sufficient considering the observed rapid temperature changes of land surfaces in some regions and during some times of the day (e.g. during morning hours)? I fear that you will get some errors because of not describing diurnal temperature changes accurately.

**Page 6, line 146:** You could consider to add a comment on how the threshold values have been chosen. Are they empirically derived or pre-calculated/pre-scribed from theory?

**Page 8, line 161**: Why the word "seeds"? Can't see the logic. It is also a bit confusing: If neighbours have similar reflectance characteristics and are labelled cloudy, why isn't the pixel itself also labelled cloudy in the first place? What is the additional information here? Or, are you looking for the case when the pixel is almost similar to its neighbours? Please explain!

**Page 8, line 187, sunglint modelling:** How accurate are those model simulations? Do they take into account wind/roughness/wave heights at sea which are very important for sunglint strength and occurrence? Thus, are you confident that the model simulations work sufficiently well?
Also, I guess you cannot rule out a sunglint-like enhancement of reflection also from low water clouds (known for having strong anisotropic reflection behaviour).

**Page 8, line 187, ratio description**: I fear that this is misleading. I mean, it seems wrong to relate the brightness temperature difference between a shortwave-infrared channel and an infrared channel to the reflectance in a shortwave channel. Shouldn't you first estimate the reflectance in the 3.9 micron channel and then calculate the ratio to the reflectance in the 0.64 micron channel? That would make more sense.

**Page 11, lines 237-238:** Difficult to understand the logics here. Stratus or fog clouds are normally trapped in a temperature inversion as a contrast to e.g., Cumulus clouds. Thus, the temperature difference to the surface may be small or even reversed (i.e., cloud top warmer than surface), especially in winter conditions over land surfaces. I fear that a lot of these clouds will be missed by these tests. Please explain!
(Maybe I misunderstand what you mean by "greater". Do you mean "warmer than" i.e., less negative in the difference?)

**Page 15, line 302:** It is a bit unclear which cloud product has really been used in the validation process. Is it one of the earlier mentioned cloud products at the 0.04 degree or 0.02 degree resolution of the cloud information. or is it products (i.e., level-2 products) with the nominal AHI pixel resolution (about 2 km)?

**Page 15, lines 304-305:** Isn't it a rather serious restriction to only select the cases when the CALIOP product is unchanged within an AHI pixel? This means that you throw away a lot of cases and only retain clouds (or clear sections) which have the scale of at least 2 or 4 km size (depending on what product you evaluate). Wouldn't it be fairer to use all cases (more representative for true conditions)? Now you don't have a clue how your schemes behave when there is sub-pixel cloudiness. For example, will these cases be considered cloudy or clear by your method and in correct proportions as given by CALIOP sub-pixel cloud information?

**Page 15, line 306**: 15 minutes is a large observation time difference. A cloud can move quite some distance in windy conditions and fast-growing cumulus clouds will not be well described. With an AHI scanning frequency of 10 minutes you should be able to at least match CALIOP within 5 minutes, shouldn't you? This would still give a large number of samples. Why was this not done instead?

**Pages 15-16, lines 309-310**: Why not CALIOP cloud tops? They should be more reliable (and more independent/objective) than any other estimation from passive imagery.

**Page 17, line 339**: I am a bit skeptical to these results since you have basically thrown away all cases with sub-pixel cloudiness and also severely underestimated the amount of thin cirrus in the CALIOP (1 km) dataset. For example, any of the investigated schemes could have misinterpreted the sub-pixel cloudy cases as either completely cloud free or completely cloudy. For example, this could have biased the estimation of all the scores in any direction. Can you give some estimation of what this restriction means in relation to the case when using all existing clouds in the CALIOP dataset? Notice that there are studies which have been able to make use of the entire CALIOP dataset (i.e., 5 km products complemented with single-shot statistics at CALIOP FOV resolution of 300 m). Two good examples are the following:

Karlsson, K.-G. and Håkansson, N.: Characterization of AVHRR global cloud detection sensitivity based on CALIPSO-CALIOP cloud optical thickness information: demonstration of results based on the CM SAF CLARA-A2 climate data record, Atmos. Meas. Tech., 11, 633–649, https://doi.org/10.5194/amt-11-633-2018, 2018.

Karlsson, K.-G.; Devasthale, A.; Eliasson, S. Global Cloudiness and Cloud Top Information from AVHRR in the 42-Year CLARA-A3 Climate Data Record Covering the Period 1979–2020. Remote Sens. 2023, 15, 3044. https://doi.org/10.3390/rs15123044

**Page 18, line 348:** Strange reasoning. Validation should always be made against reference data which are closest to the truth. To compare to another dataset based on passive imagery means that you compare with something that is very likely to suffer from the same weaknesses as your own product. Thus, results are likely to be "too good to be true". At least you should mention this and possibly state what errors can be expected. For example, MODIS product have also been evaluated against CALIOP so there is some knowledge of what differences you can expect. Notice also in the previous reference Karlsson et al., 2023 that MODIS-derived cloud tops show quite some problems when compared to CALIOP (and AVHRR) estimates.

**Page 19, line 363**: From where comes this improvement? I mean, can you point out exactly what differences in the algorithms are responsible for this improvement? It is not clear to me since your description of improvements of cloud top properties deals mainly with the cloud phase determination. Is it simply so that the JAXA algorithm has specific problems in relation to MODIS and NJIAS algorithms?

**Page 23, line 430, cloud phase results**: Again, I wonder how results would change if a larger fraction of true very thin cirrus clouds would have been present (i.e., by using the 5 km CALIOP datasets). Now only the thick and moderately thin cirrus clouds are included in the study. Can you comment this? For example, what happens in case of very thin cirrus overlying thick low-level water clouds? What phase is reported?

**Page 25, line 463**: Yes, further studies are needed to better understand the differences but these results do not need to be reported here.

**Pages 27 and 29, Figs 12 and 13**: Very nice figures showing the differences between the methods!

**Page 30, line 558**: In the summary and conclusions section, you present POD and FAR results as if they were general (i.e. for all cases). But you should state that you only studied the part of CALIOP data that showed clouds on the scale of AHI. Thus, not including effects of sub-pixel cloudiness or very thin cirrus. I think this must be commented here (and perhaps also in the abstract).

**Editorial remarks:**

**Page 1, Abstract, line 19:** Unnecessary information since Dr. Zhuge and colleagues are already listed as authors, and thus responsible for the work. Remove.

**Page 1, Abstract, line 20:** Add CALIPSO in brackets.

**Page 1, Abstract, line 20:** Add MODIS in brackets.

**Page 1, Abstract, line 24:** Remove "Then," and start sentence with "Two applications examples..."

**Page 2, line 45:** Add "CALIPSO" in brackets.

**Page 2, line 45:** Add "CPR" in brackets.

**Page 2, line 47:** Remove "often".

**Page 2, line 53:** Maybe a better formulation is "have resulted in the generation of"?

**Page 3, lines 64-68:** Add several acronyms in brackets here!

**Page 3, line 74:** Add acronym CAPCOM!

**Page 3, line 85:** I suggest to replace "clocks" with "hours".

**Page 3, line 87:** Change "objects" to "objectives".

**Page 5, table 1, CldHeight, unit**: Do you mean meters above ground level (AGL)? If so, maybe it should be explained somewhere.

**Page 6, line 131:** Change "have" to "has".

**Page 11, line 211:** Change "dusk" to "dust".

**Page 11, line 212:** Change "dusk" to "dust".

**Page 17, caption Table 5:** Explain in caption the meaning of the bold numbers (even if it can be assumed that it is the "winners").

**Page 23, line 432:** Change "Colletction" to "Collection".

**Page 23, line 440:** Write "The correlation coefficient (CC) of …."

**Page 28, line 526**: Change "Infa" to "In-Fa".

**Page 30, line 536**: I think it would not hurt (i.e., would be valuable for the reader) if you also mention which channels are used in the two RGBs.

**Page 30, line 539:** Clarify in the caption or in the text that the content of panels g-h in Figure 13 are weather radar rain rates from the Zhejiang province.

---

## Author Comment (AC1)

***Reply to the editor and referees*:**

We would like to express our sincere thanks to the editor and two referees for their detailed, constructive, and professional comments that significantly improved the manuscript. A point-by-point reply to all comments is provided below.

**a. Major revision**

1. A new figure (Figure 1 in the revised paper) is added in Section 2.3.1 to illustrate the flowchart of the NJIAS cloud mask algorithm.

2. *(Follow the suggestions of Referee 1)* In Sections 3.1-3.3, the collocation criterion in time difference between CALIOP and AHI observations is now reduced to ±5 min. By doing so, the extents of improvements upon MYD06 and JAXA are more significant than those previously using ±15 min criterion, suggesting that a shorter collocation time difference is more appropriate to the fast-growing cumulus clouds.

3. (*Follow the suggestions of Referee 1*) In Section 3.2, the cloud height parameters (CTH, CTP and CTT) are now objectively evaluated against the CALIOP 1-km cloud layer products.

4. *(Follow the suggestions of Referee 2)* Section 3.5 is newly added to highlight the key advantages and limitations of the NJIAS results compared to MYD06 and JAXA products.

**b. Minor revision**

1. Manuscript text and figure captions are edited according to the suggestions from the two referees.

**Response to** Referee 1:

Review of Earth System Science Datasets manuscript by Zhuge et al. entitled "Introduction to the NJIAS Himawari-8/9 cloud feature dataset for climate and typhoon research"

**General impression and recommendations**

This manuscript describes a new cloud climatology (NJIAS) based on data from the AHI imager onboard the Himawari 8+9 geostationary satellites. The covered region is the East Asia and North Pacific part of the Himawari disc and the temporal period covered is from 2016 to 2022. The methods for deriving various cloud parameters are described in detail. Validation results based on comparisons with data from the cloud lidar CALIOP on the CALIPSO satellite, with corresponding products from MODIS and with and operational AHI-based products from JAXA (Japan) are presented. The climatology is found to produce results compatible or, for some parameters, even better than the reference datasets from MODIS and Jaxa.

I find the study interesting because of the comprehensive use of existing methods (thus, making use of long-term international experience) but also the addition of new features which improve results even further. The manuscript is well-written, figures and tables are very appropriate and well designed, and the English language is actually surprisingly good for coming from non-native authors.

I do recommend acceptance of this manuscript but I also have a number of questions, mainly related to the methods used for validating the results. I have identified some problems with the methods and also some weaknesses in the validation methods and I think it is appropriate that authors at least discuss these weaknesses in an updated manuscript before the paper is published.

Detailed remarks and questions follow below and at the end some editorial remarks.

*Reply*: Thank you for your affirmation of this article and your constructive and detailed comments. We revised the manuscript according to your comments. The comments

have been addressed below.

**General remarks and questions:**

**Page 4, line 112:** Is the temporal resolution of 6 hours from NCEP reanalyses really sufficient considering the observed rapid temperature changes of land surfaces in some regions and during some times of the day (e.g. during morning hours)? I fear that you will get some errors because of not describing diurnal temperature changes accurately.

*Reply*: Agree. The NCEP FNL analysis with a 6-h temporal resolution, which are interpolated to AHI observation times, are insufficient for capturing the rapid changes in land surface temperatures observed in certain regions and during specific times of the day, such as early morning hours.

We revised the sentence "*The NCEP FNL analysis, which has a 0.25° × 0.25° horizontal resolution and a 6-h interval, is remapped to AHI observation times and pixels using a linear interpolation method*" in Section 2.1 (Lines 121-122).

Meanwhile, the following discussions were added in the conclusion section: "*The NJIAS HCFD is subject to uncertainties. For example, the NCEP FNL analysis with a 6-h temporal resolution, although having been interpolated to align with AHI observation times, are insufficient for capturing the rapid changes in land surface temperatures observed in certain regions and during specific times of the day, such as early morning hours. The accuracy of the fog and snow masks, which heavily depend on land surface temperature observations, could be compromised due to an inability to imprecisely represent diurnal temperature variations.*" (Lines 664-679)

**Page 6, line 146:** You could consider to add a comment on how the threshold values have been chosen. Are they empirically derived or pre-calculated/pre-scribed from theory?

*Reply*: Following your suggestion, we added in Section 2.3.1 the sentences of "*The threshold ($e$) for a certain test is generally derived via a comparison of co-located AHI/ABI with CALIOP data (Zhuge and Zou, 2016; Zhuge et al., 2017).*" (Line 158)

**Page 8, line 161**: Why the word "seeds"? Can't see the logic. It is also a bit confusing: If neighbours have similar reflectance characteristics and are labelled cloudy, why isn't the pixel itself also labelled cloudy in the first place? What is the additional information here? Or, are you looking for the case when the pixel is almost similar to its neighbours? Please explain!

*Reply*: Sorry for the misunderstanding. We added the flowchart of the NJIAS cloud mask algorithm (Figure 1 in the revised paper) and a sentence in section 2.3.1 where the RST is mentioned: "*The RST is implemented subsequent to the preliminary cloud mask determination derived from the other 14 tests*." (Line 184)

**Page 8, line 187, sunglint modelling:** How accurate are those model simulations? Do they take into account wind/roughness/wave heights at sea which are very important for sunglint strength and occurrence? Thus, are you confident that the model simulations work sufficiently well? Also, I guess you cannot rule out a sunglint-like enhancement of reflection also from low water clouds (known for having strong anisotropic reflection behaviour).

*Reply*: Yes, the surface wind speed and surface wind direction have been taken into account in the brightness temperature simulation using CRTM v2.2.3. Han et al. (2013) developed a bidirectional reflectance distribution function (BRDF) for the ocean surface. According to their assessment, the daytime biases from the BRDF model are slightly larger than nighttime biases, but the differences are within $\sim \pm 0.5$ K for all surface-sensitive shortwave channels.

As described in Zhuge and Zou (2016), the threshold for $B_{3.9\,\mu m} - O_{3.9\,\mu m} > \varepsilon_{SG\_DLS1}$ is defined as $\varepsilon_{SG\_DLS1} = -(\mu - 3\sigma)$, where $\mu$ and $\sigma$ represent the bias and standard deviation of brightness temperature differences of 3.9-μm channel between AHI observations and model simulations in clear-sky conditions. On the basis of a previous study in which the AHI data bias is estimated using CRTM (Zou et al. 2016), $\mu$ and $\sigma$ are set to 0.28 K and 0.97 K, respectively.

Of course, $B_{3.9\mu m} - O_{3.9\mu m} > \varepsilon_{SG\_DLS1}$ can't detect all the low clouds over sun-glint areas,

so we used another test ($\dfrac{\left(O_{3.9\mu m} - O_{10.4\mu m}\right)}{O_{0.64\mu m}} < \varepsilon_{SG\_DLS2}$) to compensate for it.

Chen, Y., Y. Han, P. Van Delst, and F. Weng, 2013: Assessment of shortwave infrared sea surface reflection and nonlocal thermodynamic equilibrium effects in the community radiative transfer model using IASI data. *J. Atmos. Oceanic Tech.*, **30**, 2152-2160.

**Page 8, line 187, ratio description**: I fear that this is misleading. I mean, it seems wrong to relate the brightness temperature difference between a shortwave-infrared channel and an infrared channel to the reflectance in a shortwave channel. Shouldn't you first estimate the reflectance in the 3.9 micron channel and then calculate the ratio to the reflectance in the 0.64 micron channel? That would make more sense.

*Reply*: We agree that a reflectance ratio would make more sense. The accurate radiative transfer equation for the 3.9-μm channel is

$$y = t_{ac}^{sat} \cdot B\left(T_c\right) \cdot \varepsilon_c + \left(R_{toa}^{*clr} - R_{ac}^{atm}\right) \cdot t_c^{sat} + R_{ac}^{atm}$$

$$+ f \cdot \left[ r_c + \frac{A_{sfc}}{1 - A_{sfc} \cdot A_{sph}} t_c^{sol} \cdot t_c^{sat} \right] \cdot t_{ac}^{sol} \cdot t_{ac}^{sat} , \qquad \text{(R1)}$$

where $\varepsilon_c$, $r_c$, $t_c^{sol}$, and $t_c^{sat}$ are the emissivity, reflectance, and downward and upward transmission of the cloud, $t_{ac}^{sol}$ and $t_{ac}^{sat}$ are the downward and upward atmospheric transmission above the cloud, $A_{sph}$ is the spherical albedo when the surface albedo ($A_{sfc}$) equals 0, and $f$ is a factor to transform the observed reflectance into an equivalent radiance, $B\left(T_c\right)$ is the Planck function for cloud-top temperature $T_c$, $R_{ac}^{atm}$ denotes the radiance emitted by the atmosphere above the cloud, and $R_{toa}^{*clr}$ represents the radiance received at the top of the atmosphere under clear-sky conditions,

excluding solar reflection. Eq. R1 involves too many parameters (unknown until having the CTH and DCOMP retrievals), so we have ever tried to estimate the 3.9-µm reflectance by simply using the formula $r_c = \dfrac{B(O_{3.9\,\mu m}) - B(O_{10.4\,\mu m})}{f - B(O_{10.4\,\mu m})}$, assuming that $r_c + \varepsilon_c = 1$, as well as $t_c^{sol}$, $t_c^{sat}$, $t_{ac}^{sol}$ and $t_{ac}^{sat}$ all equal to 1. However, the $r_c$ value we calculated was unbelievable, because the assumption $r_c + \varepsilon_c = 1$ does not hold true over the sun-glint areas!

In contrast, it is more effective when simply using the formula $\dfrac{\left(O_{3.9\,\mu m} - O_{10.4\,\mu m}\right)}{O_{0.64\,\mu m}}$ to identify clouds over the sun-glint areas.

**Page 11, lines 237-238:** Difficult to understand the logics here. Stratus or fog clouds are normally trapped in a temperature inversion as a contrast to e.g., Cumulus clouds. Thus, the temperature difference to the surface may be small or even reversed (i.e., cloud top warmer than surface), especially in winter conditions over land surfaces. I fear that a lot of these clouds will be missed by these tests. Please explain!

(Maybe I misunderstand what you mean by "greater". Do you mean "warmer than" i.e., less negative in the difference?)

*Reply*: Sorry for it, I meant "warmer than". Now the words "greater than" have been changed to "less negative than" (Line 257).

**Page 15, line 302:** It is a bit unclear which cloud product has really been used in the validation process. Is it one of the earlier mentioned cloud products at the 0.04 degree or 0.02 degree resolution of the cloud information, or is it products (i.e., level-2 products) with the nominal AHI pixel resolution (about 2 km)?

*Reply*: It is at the nominal 2-km pixel level. The sentence is revised as follows: "…*are objectively evaluated at the **nominal 2-km** pixel level against the CALIOP 1-km cloud layer products of version 4.20 (Avery et al., 2020) in the whole year of 2017*" (Line 324).

Besides, the raw cloud product named FLDK (for Segments 2–4 of the full disk imagery) is added in Table 4 for clarity.

**Page 15, lines 304-305:** Isn't it a rather serious restriction to only select the cases when the CALIOP product is unchanged within an AHI pixel? This means that you throw away a lot of cases and only retain clouds (or clear sections) which have the scale of at least 2 or 4 km size (depending on what product you evaluate). Wouldn't it be fairer to use all cases (more representative for true conditions)? Now you don't have a clue how your schemes behave when there is sub-pixel cloudiness. For example, will these cases be considered cloudy or clear by your method and in correct proportions as given by CALIOP sub-pixel cloud information?

*Reply*: Thank you for this comment. In the revised paper, we added the statistic result for the sub-pixel cloudy cases. Accordingly, Figure 5 (Fig. 4 in the submitted paper before revision) is redrawn. These sentences are added: "*Three products (MYD06, NJIAS and JAXA) have a probability of 25–35% to classify sub-pixel cloudy cases as confidently clear or probably clear over oceans or during daytime. This probability increases to approximately 47% for the NJIAS product over continental areas at night.*" (Lines 346-349)

**Page 15, line 306**: 15 minutes is a large observation time difference. A cloud can move quite some distance in windy conditions and fast-growing cumulus clouds will not be well described. With an AHI scanning frequency of 10 minutes you should be able to at least match CALIOP within 5 minutes, shouldn't you? This would still give a large number of samples. Why was this not done instead?

*Reply*: In Sections 3.1-3.3, the temporal difference between CALIOP and AHI observations is now reduced to ±5 min. The extents of improvements upon MYD06 and JAXA are more significant than those using temporal window of ±15 min, which agrees with your suggestion that a shorter collocation time difference is more appropriate to the fast-growing cumulus clouds.

**Pages 15-16, lines 309-310**: Why not CALIOP cloud tops? They should be more reliable (and more independent/objective) than any other estimation from passive imagery.

*Reply*: We accepted this suggestion. In Section 3.2, the cloud height parameters (CTH, CTP and CTT) are now objectively evaluated against the CALIOP 1-km cloud layer products. Section 3.2 has been largely rewritten. Here, the sentence has changed to "*In this section, results obtained by the NJIAS cloud mask and **cloud-top property** algorithms are objectively evaluated…*" (Lines 323-324)

**Page 17, line 339**: I am a bit skeptical to these results since you have basically thrown away all cases with sub-pixel cloudiness and also severely underestimated the amount of thin cirrus in the CALIOP (1 km) dataset. For example, any of the investigated schemes could have misinterpreted the sub-pixel cloudy cases as either completely cloud free or completely cloudy. For example, this could have biased the estimation of all the scores in any direction. Can you give some estimation of what this restriction means in relation to the case when using all existing clouds in the CALIOP dataset? Notice that there are studies which have been able to make use of the entire CALIOP dataset (i.e., 5 km products complemented with single-shot statistics at CALIOP FOV resolution of 300 m). Two good examples are the following:

Karlsson, K.-G., Devasthale, A.,and Eliasson, S: Global Cloudiness and Cloud Top Information from AVHRR in the 42-Year CLARA-A3 Climate Data Record Covering the Period 1979–2020, Remote Sens., 15, 3044. https://doi.org/10.3390/rs15123044, 2023.

Karlsson, K.-G. and Håkansson, N.: Characterization of AVHRR global cloud detection sensitivity based on CALIPSO-CALIOP cloud optical thickness information: demonstration of results based on the CM SAF CLARA-A2 climate data record, Atmos. Meas. Tech., 11, 633–649, https://doi.org/10.5194/amt-11-633-2018, 2018.

*Reply*: We accepted this suggestion. The following sentences have been added: "*Note that the aforementioned statistical analysis excluded all cases with sub-pixel cloudiness or very thin cirrus (Karlsson et al., 2018; 2023). If the sub-pixel cloudy cases were*

*misinterpreted as either completely clear-sky or completely cloudy, the estimation of all the scores would be biased unpredictably.*" (Lines 368-371)

**Page 18, line 348:** Strange reasoning. Validation should always be made against reference data which are closest to the truth. To compare to another dataset based on passive imagery means that you compare with something that is very likely to suffer from the same weaknesses as your own product. Thus, results are likely to be "too good to be true". At least you should mention this and possibly state what errors can be expected. For example, MODIS product have also been evaluated against CALIOP so there is some knowledge of what differences you can expect. Notice also in the previous reference Karlsson et al., 2023 that MODIS-derived cloud tops show quite some problems when compared to CALIOP (and AVHRR) estimates.

*Reply*: Agree again. In Section 3.2, the cloud height parameters (CTH, CTP and CTT) are now objectively evaluated against the CALIOP 1-km cloud layer products. Section 3.2 has been rewritten.

**Page 19, line 363**: From where comes this improvement? I mean, can you point out exactly what differences in the algorithms are responsible for this improvement? It is not clear to me since your description of improvements of cloud top properties deals mainly with the cloud phase determination. Is it simply so that the JAXA algorithm has specific problems in relation to MODIS and NJIAS algorithms?

*Reply*: It is because that JAXA employs a simple conventional methodology. The following sentences have been added to offer some explanations to this improvement: "*The JAXA operational cloud height algorithm incorporates the IR window technique, the radiance rationing technique, and the IR-water vapor intercept technique, and choose one of them contingent upon the result of cloud type classifications (Mouri et al., 2016b). This conventional methodology is different from the maximum likelihood estimation algorithms, such as the ACHA.*" (Lines 399-402)

**Page 23, line 430, cloud phase results**: Again, I wonder how results would change if

a larger fraction of true very thin cirrus clouds would have been present (i.e., by using the 5 km CALIOP datasets). Now only the thick and moderately thin cirrus clouds are included in the study. Can you comment this? For example, what happens in case of very thin cirrus overlying thick low-level water clouds? What phase is reported?

*Reply*: A subsection (Section 3.5) is added to highlight the key advantages and limitations of the NJIAS results compared to MYD06 and JAXA products. Seen from the added Fig. 13, both the MYD06 and NJIAS products demonstrate good performances in multilayer cloud cases, and report an ice phase in region "C" where thin cirrus clouds were overlying low-level water clouds (Figs. 13f and 13g). In contrast, the JAXA product gives a liquid-water phase in region "C" (Fig. 13h), suggesting that the JAXA cloud-top phase algorithm requires further enhancement.

**Page 25, line 463**: Yes, further studies are needed to better understand the differences but these results do not need to be reported here.

*Reply*: Thank you.

**Pages 27 and 29, Figs 12 and 13**: Very nice figures showing the differences between the methods!

*Reply*: Thank you for your affirmation and encouragement.

**Page 30, line 558**: In the summary and conclusions section, you present POD and FAR results as if they were general (i.e. for all cases). But you should state that you only studied the part of CALIOP data that showed clouds on the scale of AHI. Thus, not including effects of sub-pixel cloudiness or very thin cirrus. I think this must be commented here (and perhaps also in the abstract).

*Reply*: We accepted this suggestion. In the abstract, this sentence was added: "*All evaluations are performed at the nominal 2 km scale, not including the effects of sub-pixel cloudiness or very thin cirrus.*" (Lines 25-26). In the summary and conclusion section, this sentence was added: "*Note that in the statistical analysis, CALIOP cases with sub-pixel cloudiness or very thin cirrus (Karlsson et al., 2018; 2023) have been*

*excluded.*" (Lines 648-650)

**Editorial remarks:**

**Page 1, Abstract, line 19:** Unnecessary information since Dr. Zhuge and colleagues are already listed as authors, and thus responsible for the work. Remove.

*Reply*: The suggested edit is incorporated into the revised manuscript.

**Page 1, Abstract, line 20:** Add CALIPSO in brackets.

*Reply*: Done.

**Page 1, Abstract, line 20:** Add MODIS in brackets.

*Reply*: Done.

**Page 1, Abstract, line 24:** Remove "Then," and start sentence with "Two applications examples..."

*Reply*: Done.

**Page 2, line 45:** Add "CALIPSO" in brackets.

*Reply*: Done.

**Page 2, line 45:** Add "CPR" in brackets.

*Reply*: Done.

**Page 2, line 47:** Remove "often".

*Reply*: Done.

**Page 2, line 53:** Maybe a better formulation is "have resulted in the generation of"?

*Reply*: Accept.

**Page 3, lines 64-68:** Add several acronyms in brackets here!

*Reply*: Done.

**Page 3, line 74:** Add acronym CAPCOM!

*Reply*: Done.

**Page 3, line 85:** I suggest to replace "clocks" with "hours".

*Reply*: The words "*are generated at full and half clocks*" have been revised as "*are generated at the 0.5 h interval*".

**Page 3, line 87:** Change "objects" to "objectives".

*Reply*: Done.

**Page 5, table 1, CldHeight, unit**: Do you mean meters above ground level (AGL)? If so, maybe it should be explained somewhere.

*Reply*: Accept. This sentence has been modified: "*Besides, the CTH in the NJIAS algorithm is measured above ground level (AGL), i.e., true altitude minus terrain elevation, which is different from the definition used in the MYD06 algorithm and the ACHA.*" (Line 244-246)

**Page 6, line 131:** Change "have" to "has".

*Reply*: We didn't change it, because "improvements" is plural. (Lines 141-142)

**Page 11, line 211:** Change "dusk" to "dust".

*Reply*: Done. Sorry for this mistake.

**Page 11, line 212:** Change "dusk" to "dust".

*Reply*: Done. Sorry for this mistake.

**Page 17, caption Table 5:** Explain in caption the meaning of the bold numbers (even if it can be assumed that it is the "winners").

*Reply*: Yes, we added a sentence to explain the meaning of the bold numbers: "*The highest skill scores for each scenario are shown in boldface*." (Line 375)

**Page 23, line 432:** Change "Colletction" to "Collection".

*Reply*: Done. Sorry for this mistake.

**Page 23, line 440:** Write "The correlation coefficient (CC) of …."

*Reply*: "CC" has been defined earlier in the text (See Line 386).

**Page 28, line 526:** Change "Infa" to "In-Fa".

*Reply*: Done. Sorry for this mistake.

**Page 30, line 536:** I think it would not hurt (i.e., would be valuable for the reader) if you also mention which channels are used in the two RGBs.

*Reply*: Accept. The caption has been revised as "*(a–b) AHI TC-RGB composite images, as outlined in Chen et al. (2022), featuring two modes with distinct color representations: (a) for the day mode (red, 0.64 μm; green, 0.64 μm; blue, 11.2 μm reversed), cirrus appears blue, convective clouds appear white, and low clouds appear yellow; and (b) for the night mode (red, 12.3 μm -10.4 μm; green, 10.4 μm - 3.9 μm; blue, 11.2 μm reversed), cirrus appears blue, low clouds appear bright green, and convective clouds appear dark violet…*"(Lines 622-626)

**Page 30, line 539**: Clarify in the caption or in the text that the content of panels g-h in Figure 13 are weather radar rain rates from the Zhejiang province.

*Reply*: The rain rates were measured by >500 gauges in Zhejiang Province. The sentence has been changed to "*The rain rates measured by rain gauges were…*" (Line 609) and the caption has been revised as "…*maximum **gauge** rain rate within*…".(Line 629)

**Response to** Referee 2:

The manuscript by Zhuge et al. presents a comprehensive introduction and evaluation of their NJIAS cloud product based on Himawari-8/9 measurements over East Asia and the western North Pacific, and the dataset is expected to serve climate and typhoon studies over the region. The cloud properties are compared with both active and passive satellite results of the kinds, and are evaluated by comparing with Himawari operational and MODIS results. The results are well organized and presented, and the manuscript is well-written overall. I have several suggestions to further strengthen the manuscript.

1. The introduction could be expanded to better motivate the need for this new dataset. How does it build upon and improve existing Himawari-8/9 cloud products? What new capabilities does it offer? More background on current limitations and gaps would help frame the value addition.

*Reply*: The introduction is expanded by adding the following sentences in the introduction section:

"*As a result, only the semi-diurnal variation of cloud cover (e.g., Shang et al., 2018; Yu et al., 2022) or convective activity (e.g., Li et al., 2021) during the daytime can be obtained from the AHI level-2 operational cloud product.*"(Lines 82-84)

"*Over the past three years, it has been discovered that the NJIAS cloud retrieval algorithms have several shortcomings and weaknesses, such as inadequate detection of low-level clouds at high solar zenith angles or over snow-covered surfaces, and insufficient masks of dust, haze and fog. Accordingly, a number of enhancements to the NJIAS cloud retrieval algorithms have been implemented. Finally, 30 variables are generated at the 0.5 h interval in the 7 yr period from April 2016 to December 2022 using these algorithms.*" (Lines 90-95)

2. In Section 2.3 on algorithm updates, more quantitative details could be provided on the impact of the refinements. For example, metrics like changes in POD, FAR, etc.

would strengthen this section.

*Reply*: Providing skill scores for the algorithm performance before and after enhancements would strengthen this section. However, implementing this idea presents certain challenges. For example, a significant improvement in the NJIAS cloud mask algorithm is its effective detection of low-level clouds at high solar zenith angles. Yet, we cannot quantitatively evaluate this because the "truth" data from MODIS or CALIOP are collected from the late-morning (9:30-10:00 LST) or afternoon (13:30-14:00 LST) orbits. The MODIS or CALIOP cloud products cannot provide samples with high solar zenith angles.

3. The evaluation results are thorough, but the authors could better highlight one or two key advantages and limitations of the NJIAS results compared to MYD06 and JAXA products.

*Reply*: In responding to this suggestion, a case study (Section 3.5) is added to highlight the key advantages and limitations of the NJIAS results compared to MYD06 and JAXA products.

4. The manuscript indicates that the dataset can be used for climate studies, while 7-year observations will be difficult to be used for climate studies. Thus, the corresponding discussions are suggested to be more carefully given.

*Reply*: We agree with this comment. The term "climate", when used in a more restricted sense, means the weather averaged over a very long period, typically 30 years. However, climate in a wider sense is the state, including a statistical description, of the earth system over a period of time, ranging from months to thousands or millions of years, which is in accordance with the following definition given by the JMA (https://ds.data.jma.go.jp/tcc/tcc/library/library2021/lectures/3_Introduction_to_Clima tology_hosaka_20211207.pdf): *"Climate, sometimes understood as the "average weather," is defined as the measurement of the mean and variability of relevant quantities of certain variables (such as temperature, precipitation or wind) over a period of time, ranging from months to thousands or millions of years"*.

5. In Section 2.4 on the cloud products, more details should be provided on the algorithms and methodology used to derive the 0.04Deg and TyWNP products from the full disk data. How was the remapping done? What is the impact on data quality compared to full resolution?

*Reply*: We accepted this suggestion. These sentences were added in Section 2.4:

 "*Currently, the NJIAS HCFD has three cloud products, namely **FLDK (for Segments 2–4 of the full disk imagery),** 0.04Deg (on regular latitude-longitude grids at 0.04 °× 0.04 °resolution) and TyWNP (for WNP Typhoons). The 0.04Deg and TyWNP products can be directly derived from the FLDK product **via projection conversion using the nearest-neighbor approach**.*" (Lines 311-314)

"*A finer resolution would retain more clouds of ~2 km size.*" (Lines 316-317)

To illustrate how to implement the projection conversion, a MATLAB code example is given here. Users can log on anonymously ftp://222.190.246.206 (port: 40028) and access the NJIAS_HCFD–FLDK product under the directory */NJIAS_HCFD_v1/FLDK/* and the MATLAB function *Him_lonlat_to_pixlin.m* under the directory */NJIAS_HCFD_v1/FLDK/Supp/*.

```
Lon = 90:0.04:190; lat = 50:-0.04:10;
[Mlon, Mlat] = meshgrid(lon,lat);
[Mpix, Mlin] = Him_lonlat_to_pixlin(Mlon',Mlat');
Mlin(Mlin>2200|Mlin<1) = 1;Mpix(Mpix>5500|Mpix<1) = 1;
Ind1 = sub2ind([5500,5500],uint32(round(Mpix)),uint32(round(Mlin)));
clear Mlat Mlon Mpix Mlin;

lin = ncread(ncfilename, 'line');
pix = ncread(ncfilename, 'pixel');
cldtype(pix, lin) = ncread(ncfilename, 'CldType');

figure(1)
imagesc(pix, lin, cldtype'); % full-disk
figure(2)
newcldtype = cldtype(Ind1);
imagesc(lon, lat, newcldtype'); % Equal-Longitude-Latitude grid
```

6. How is the uncertainties of the dataset, is there any estimation?

*Reply*: In the revised paper, we added discussions about the uncertainties. The

quantitative assessment of the uncertainties associated with the NJIAS HCFD will be the focus of future investigations.

The following discussions were added in the conclusion section: "*The NJIAS HCFD is subject to uncertainties. For example, the NCEP FNL analysis with a 6-h temporal resolution, although having been interpolated to align with AHI observation times, are insufficient for capturing the rapid changes in land surface temperatures observed in certain regions and during specific times of the day, such as early morning hours. The accuracy of the fog and snow masks, which heavily depend on land surface temperature observations, could be compromised due to an inability to imprecisely represent diurnal temperature variations. Furthermore, given the systematic overestimation found in the NJIAS $Re_{2.3}$ retrieval, an in-depth inter-sensor radiometric analysis is crucial. A radiometric adjustment factor, which excludes the effect of central wavelength shift, can be employed for aligning AHI's relative radiometric calibration more closely with that of the MODIS. The quantitative assessment of the uncertainties associated with the NJIAS HCFD will be the focus of future investigations.*" (Lines 664-674)

7. The intercomparison between NJIAS and JAXA products reveals some interesting differences in Section 3, but the sources of discrepancies are not thoroughly discussed. Speculating on the potential reasons behind the weaker performance of JAXA and areas for algorithm improvement would add value.

*Reply*: We accepted this suggestion. The following sentences were added to discuss the sources of discrepancies:

"*Besides, the JAXA product classifies some clear-sky pixels and a majority of cloudy pixels as probably cloudy over the sun-glint areas (Fig. 13e). This is the reason for JAXA dataset to have high PODs but also high FARs.*" (Lines 533-535)

"*It is noteworthy that the NJIAS retrievals tend to slightly underestimate CTH and overestimate both CTP and CTT for high clouds, possibly due to the fact that only a single channel centered at 13.3 μm is allocated within the broad carbon dioxide absorption region for the AHI.*" (Lines 388-390)

"*Incorporating additional carbon dioxide absorption channels would enhance the*

*inference of cloud-top pressure and effective cloud amount for high-level clouds, especially semi-transparent clouds such as cirrus (Platnick et al., 2019).*" (Lines 392-394)

"*The JAXA operational cloud height algorithm incorporates the IR window technique, the radiance rationing technique, and the IR-water vapor intercept technique, and choose one of them contingent upon the result of cloud type classifications (Mouri et al., 2016b). This conventional methodology is different from the maximum likelihood estimation algorithms, such as the ACHA.*" (Lines 399-402)

"*Meanwhile, the MYD06 SWIR+IR retrievals (Fig. 9) show a significant improvement over the IR-only retrievals (Fig. 8) by supplementing the IR tests with those from solar channels.*" (Lines 462-464)

"*The overestimations are likely due to a discrepancy in the sensor central wavelengths which will affect the reflectance observations and the DCOMP LUTs (Wang et al., 2018). Interestingly, the overestimations are not found in the JAXA retrievals. A detailed comparison of the LUTs used by the NJIAS and the JAXA is essential.*" (Lines 505-508)

A case study is also added in Section 3.5 to highlight the key advantages and limitations of the NJIAS results compared to MYD06 and JAXA products. The sources of discrepancies for the case study were discussed in Section 3.5.

8. The conclusions in Section 6 are a bit abrupt and underdeveloped. The authors could expand on the key findings, significance, broader applications and future directions to improve this section. Reiterating the unique contributions of this work would be beneficial.

*Reply*: We accepted this suggestion. The following actions were taken in responding to the comment:

(1) In the revised paper, the key findings are rewritten as:

"*1) The POD and FAR of the NJIAS HCFD for cloud detections are ~88% and ~6%, respectively. The NJIAS HCFD gives higher skill scores than the MYD06 during nighttime. For daytime scenario, the NJIAS HCFD lags behind the MYD06, but*

*outperforms JAXA dataset. Note that in the statistical analysis, CALIOP cases with sub-pixel cloudiness or very thin cirrus (Karlsson et al., 2018; 2023) have been excluded.*

*2) The three cloud height parameters (CTT, CTH and CTP) derived from the NJIAS HCFD show better agreement with the CALIOP data than those obtained from the MYD06. The NJIAS retrievals tend to slightly underestimate CTH and overestimate both CTP and CTT for high clouds. The JAXA product has a more pronounced tendency to underestimate the CTH and overestimate the CTT of mid-to-high-level clouds.*

*3) The PODs of the NJIAS phase determinations for the CALIOP liquid-water and ROI cloud tops are 82.60% (82.17%) and 88.59% (85.35) over oceans (land), respectively. Problems are found for the MYD06 and JAXA retrievals, such as misclassifying pixels with a CTT greater than 0 ℃ as ice phase over ocean, and misclassifying pixels with a CTT below -40 ℃ as non-ice phase over land.*

*4) Overall, the NJIAS DCOMP retrievals have high correlations with the Collection-6.1 MYD06 results, with CC ranging from 0.722 to 0.853. The JAXA dataset only provides Re values retrieved from the AHI 2.3-μm channel. However, the overestimation in the NJIAS Re2.3 retrieval is not found in the JAXA retrievals.*" (Lines 646-663)"

(2) We also added some discussions about the uncertainties to the conclusion section: "*The NJIAS HCFD is subject to uncertainties. For example, the NCEP FNL analysis with a 6-h temporal resolution, although having been interpolated to align with AHI observation times, are insufficient for capturing the rapid changes in land surface temperatures observed in certain regions and during specific times of the day, such as early morning hours. The accuracy of the fog and snow masks, which heavily depend on land surface temperature observations, could be compromised due to an inability to imprecisely represent diurnal temperature variations. Furthermore, given the systematic overestimation found in the NJIAS Re2.3 retrieval, an in-depth inter-sensor radiometric analysis is crucial. A radiometric adjustment factor, which excludes the effect of central wavelength shift, can be employed for aligning AHI's relative radiometric calibration more closely with that of the MODIS. The quantitative*

*assessment of the uncertainties associated with the NJIAS HCFD will be the focus of future investigations.*" (Lines 664-674)